



# Vertical segregation among pathways mediating nitrogen-loss (N₂ and N₂O production) across the oxygen gradient in a coastal upwelling ecosystem

Alexander Galán,[1,a] Bo Thamdrup,[2] Gonzalo S. Saldías,[3,4] and Laura Farías[5,6,7*]

[1] CREA – Centro Regional de Estudios Ambientales, Universidad Católica de la Santísima Concepción, Av. Colón 2766, Talcahuano 4270789, Chile. [a]Formerly at: Departamento de Oceanografía, Universidad de Concepción, Concepción, Chile.
[2] Department of Biology and Nordic Center for Earth Evolution (NordCEE), University of Southern Denmark, Odense M, Denmark
[3] College of Earth, Ocean, and Atmospheric Sciences, Oregon State University, Corvallis, USA.
[4] Centro FONDAP de Investigación en Dinámica de Ecosistemas Marinos de Altas Latitudes (IDEAL), Valdivia, Chile
[5] Departamento de Oceanografía, Universidad de Concepción, Concepción, Chile
[6] Laboratorio de Procesos Oceanográficos y Clima (PROFC), Universidad de Concepción, Concepción, Chile
[7] Centro de Ciencia del Clima y la Resiliencia (CR2), Chile

*Correspondence to*: Laura Farías (laura.farias@udec.cl)



**Abstract.** The upwelling system off central Chile (36.5° S) is seasonally subjected to oxygen ($O_2$) deficient waters, with a strong vertical gradient in $O_2$ varying from oxic to anoxic conditions on a scale of a few meters (30-50 m interval) over the shelf. This condition inhibits and/or stimulates processes involved in nitrogen (N) removal (e.g., anammox, denitrification, and nitrification). During austral spring (September 2013) and summer (January 2014), the main pathways involved in N-loss and its speciation, in the form of $N_2$ and $N_2O$, were studied using $^{15}N$-tracer incubations, inhibitor assays, the natural abundance of nitrate isotopes, and hydrographic information. Incubations were developed using water retrieved from the oxycline (25 m depth) and bottom waters (85 m depth), over the continental shelf off Concepción, Chile. Results of $^{15}N$-labeled incubations revealed a higher N removal activity during the austral summer, with denitrification as the dominant $N_2$ producing pathway, which occurred together with anammox at all times. Interestingly, in both spring and summer maximum potential N removal rates were observed in the oxycline. Notwithstanding, a greater availability of oxygen was observed (maximum $O_2$ fluctuation between 270 and 40 µmol $L^{-1}$), relative to the hypoxic bottom waters (< 20 µmol $O_2$ $L^{-1}$). Different pathways were responsible for $N_2O$ produced in the oxycline and bottom waters, with ammonium oxidation and dissimilatory nitrite reduction, respectively, as the main subsidiary processes. Ammonium produced by DNRA could sustain both anammox and nitrification rates, including the ammonium utilized for $N_2O$ production. The temporal and vertical variability of $\delta^{15}N$-$NO_3^-$ confirms that multiple N-cycling processes are modulating the isotopic nitrate composition over the shelf off central Chile during spring and summer. N removal processes in this coastal system appear to be related to the availability and distribution of oxygen and particles, which are a source of organic matter and the fuel for the production of other electron donors (i.e., ammonium) and acceptors (i.e., nitrate and nitrite), after its remineralization. These results highlight the links between several pathways involved in N-loss. They also establish that different mechanisms, supported by alternative N substrates, are responsible for a substantial accumulation of $N_2O$, frequently observed as hotspots in the oxycline and bottom waters. Considering the extreme variation in oxygen observed in several coastal upwelling systems, these findings could help to understand the ecological and biogeochemical implications due global warming where intensification and/or expansion of the oceanic OMZs is projected.





## 1 Introduction

It is widely accepted that fixed nitrogen (N) availability influences marine primary productivity (Falkowski et al., 1998). Therefore, fixed-N loss from the ocean has an important influence on ecosystem functioning and global biogeochemical cycles. A significant fraction of the global fixed-N removal (30-50 %) occurs in

oxygen minimum zones (OMZs), where dissolved oxygen (DO) levels as low as 2–4 µmol $L^{-1}$ or lower activate nitrate-based anaerobic metabolisms (Devol, 1978; Dalsgaard et al., 2014). Fixed-N removal occurs predominantly in the form of $N_2$, generated by canonical denitrification and anammox processes (Gruber and Sarmiento, 1997; Codispoti et al., 2001; Devol, 2003). The steep oxygen ($O_2$) gradients observed at the OMZ's boundaries further trigger a substantial production and accumulation of nitrous oxide - $N_2O$ (Naqvi et

al., 2010; Dalsgaard et al., 2012), which may eventually be emitted to the atmosphere, thus contributing to the overall N loss. OMZs are notable features of eastern boundary ecosystems (i.e., South and North Pacific, South Atlantic), and the Arabian Sea, where seasonal or permanent wind-driven upwelling sustains remarkably high levels of biological production, and subsequently a high respiration rate from subsurface organic matter (Helly and Levin, 2004).

The Eastern South Pacific (ESP) is influenced by the shallow poleward Peru-Chile undercurrent (e.g., Blanco et al., 2001), which transports nutrient-rich, high-saline, and $O_2$-depleted Equatorial Subsurface Waters – ESSW (Strub et al., 1998). During austral spring-summer (September-April), a marked upwelling-favorable period driven by persistent southwesterly winds, ESSW arrive to the coastal area off central Chile impinging

on one of the widest continental shelves in Chile (Sobarzo and Djurfeldt, 2004; Sobarzo et al., 2007). Throughout the upwelling season high primary production rates (Daneri et al., 2000) occur along with a large downward flux of organic matter, leading to an increase in respiration rates at depth (Montero et al., 2007). These conditions, coupled with the presence of already $O_2$-deficient waters, and greater availability of electron donors and acceptors (as a result of the advection of inorganic substrates, local organic matter

respiration, and autotrophic redox reactions), create steep gradients for $O_2$ and other chemicals, with the establishment of hypoxic-anoxic conditions at middle-bottom depths. This leads to an active removal of fixed N and greenhouse gas production (Farías et al., 2009; Galán et al., 2014). Likewise, the dense upwelled waters cause increased stratification in the water column, decelerating the downward flux of organic particles formed at the surface (Charpentier et al., 2007), facilitating their accumulation, and thus creating anoxic

microbial hotspots within the oxycline that favor the N-based anaerobic microbial metabolisms (Klawonn et al., 2015; Stief et al., 2016).

As previously mentioned, OMZs are also characterized by sustaining an important production and outgassing of $N_2O$ (e.g., Nevison et al., 2004, Naqvi et al., 2010), which is clearly evident in this system (Farías et al.,

2015). $N_2O$ is a potent greenhouse gas, with a ~300 times greater radiative effect than $CO_2$, and also contributing to the destruction of the stratospheric ozone (Ravishankara et al., 2009, Portman et al., 2012).





The distribution and accumulation of $N_2O$ is highly sensitive to $O_2$ levels, and thus far both heterotrophic and autotrophic processes are known to contribute to their cycling (Ritchie and Nicholas, 1972; Wrage et al., 2001; Codispoti, 2010). Nevertheless, the mechanism or mechanisms responsible for $N_2O$ accumulation in both the upper and lower OMZ oxyclines remain unclear. Under anoxia, $N_2O$ produced is almost completely reduced to $N_2$. This reduction is, however, inhibited with increasing $O_2$ levels, and $N_2O$ accumulates and in some cases becomes the main product of denitrification processes (Kester et al., 1977; Goreau et al., 1980; Bonin et al., 1989; but see also Qu et al., 2016). $N_2O$ can also be produced during ammonium ($NH_4^+$) oxidation, a chemolithoautotrophic process by which $NH_4^+$ is oxidized aerobically either by bacteria or archaea. However, the pathways of $N_2O$ production appear to differ between ammonium oxidizers from these two domains (Blainey et al., 2011; Kim et al., 2011; Spang et al., 2012; Kozlowski et al., 2016). While $N_2O$ is a minor product in ammonium oxidation (up to 10 % relative to $NO_2^-$ production; Goreau et al., 1980), the yield of $N_2O$ produced by nitrifiers relative to $NO_2^-$ increases ~20 times as $O_2$ saturation in the atmosphere diminishes to 1% (Kester et al., 1977). $N_2O$ is formed as a byproduct from hydroxylamine ($NH_2OH$) and nitric oxide (NO) precursors in ammonium oxidizing bacteria (Wrage et al., 2001; Arp and Stein, 2003; Stein, 2011), and during situations of $O_2$ stress, this bacterial group could potentially reduce $NO_2^-$ produced into $N_2O$ via NO, through nitrifier-denitrification process (Ritchie and Nicholas, 1972; Poth and Focht, 1985; Shaw et al., 2006). Conversely, and despite several reports suggested that archaeal ammonium oxidizers are an important source of $N_2O$ based on their ubiquity, abundance, activity and high $NH_3$ affinity (Könneke et al., 2005, Wuchter et al., 2006, Martens-Habbena et al., 2009, Santoro et al., 2011; Jung et al., 2014), the pathway through which the $N_2O$ is produced by this group is still not fully understood. Nonetheless, it was recently demonstrated that Archaea involved in ammonium oxidation do not have the enzymatic capacity to reduce NO to $N_2O$ through nitrifier-denitrification and $N_2O$ formation was suggested to occur through a hybrid pathway pairing nitrogen from $NH_2OH$ and NO (Stieglmeier et al., 2014; Kozlowski et al., 2016).

Although the atmospheric $N_2O$ concentration is increasing by ~ 0.25 % annually (IPCC AR5, 2014), estimates of the marine contribution to this are highly uncertain. This is mainly due to the lack of coastal upwelling areas included in the global estimates of oceanic $N_2O$ emissions, although the contribution from these regions is potentially significant (Bange et al., 1996; Nevison et al., 2004). While several decades of research have focused on measuring (or estimating) the products of N loss (as $N_2$ and $N_2O$), few studies have tracked the sources to determine the dominant process, or to establish if loss products are formed as the result of coupling among multiple N cycling pathways. Despite both denitrification and anammox being considered as major loss terms for available N (as either $N_2$ and/or $N_2O$), canonical nitrification or nitrifier denitrification may also contribute to removal in systems with high $O_2$ variability. Thus, when low $O_2$ concentrations prevail, it is possible that multiple microbial N loss processes operate simultaneously. However, the relative contribution of each metabolic pathway, and direct and indirect processes involved in N loss, remains unsolved. This study investigates variation in the N-based metabolisms of the microbial community, coupled with N loss, in the seasonally stratified and $O_2$-limited waters of the coastal upwelling system off central Chile, during the spring transition and summer, across a gradient from oxic to anoxic conditions. We used a





combination of $^{15}N$ labeled activity measurements, inhibitory assays, and analysis of natural nitrate isotope abundances. The results obtained from this coastal ecosystem, where extreme $O_2$ variability is seasonally observed, could help to understand the ecological and biogeochemical implications of global warming, which is expected to cause intensification and expansion of oceanic OMZs (Keeling et al., 2010; Deutsch et al., 2011; Helm et al., 2011).

## 2 Methods

### 2.1 Study area and sampling strategy

The study was carried out at the COPAS Time Series station 18 (TS Sta. 18 - 36°30' S; 73°08' W, http://www.copas.cl, see Escribano and Schneider, 2007) located ~ 33 km from the coast over the middle continental shelf (~ 92 m isobath) off central Chile (Fig. 1). Sampling was conducted during austral spring transition (12 September 2013) and summer (28 January 2014) on board the RV *Kay Kay II*. Continuous hydrographic profiles (temperature, salinity, $O_2$) were obtained using a conductivity–temperature–depth (CTD) device outfitted with an $O_2$ sensor (Seabird 23B; accuracy at 2 % of saturation: □~ 2 μmol $O_2$ L□$^1$). Discrete seawater samples (5-10, 25, 40-60, and 80-85 m depths) were collected with 10 L Niskin bottles attached to a rosette sampler for hydrographic measurements (dissolved oxygen [DO]; N species - $NH_4^+$, $NO_3^-$, $NO_2^-$, $N_2O$; and $PO_4^{3-}$). A laser sensor (LISST-25X) device was used to measure the abundance, size (mean diameter), and continuous vertical distribution of suspended particles.

### 2.2 Hydrographic analyses

Seawater samples for discrete hydrographic measurements were taken in triplicate. Filtered (0.7 μm, GF/F glass fiber filter) water was analyzed for $NO_3^-$, $NO_2^-$ and $PO_4^{3-}$ using standard colorimetric methods (Grasshoff et al., 1983) in an automatic analyzer (Seal Analytical). $NH_4^+$ was measured fluorometrically from fresh samples (40 mL), using the orthopthaldialdehyde method (Holmes et al., 1999) on a Turner Design Trilogy fluorometer (Turner Design). The relative standard error for $NO_3^-$, $NO_2^-$, $PO_4^{3-}$ and $NH_4^+$ was lower than ± 10 %, ± 3 %, ± 3 %, and ± 5 %, respectively. $N_2O$ was analyzed from the equilibrated headspace (5 mL) in 20 mL crimp-cap bottles by gas chromatography (Varian 3380) using an electron capture detector maintained at 350°C.

### 2.3 Satellite-derived chlorophyll-a

High resolution (1 km) MODIS (Moderate Resolution Imaging Spectroradiometer) imagery of Chlorophyll-*a* (Chl-*a*) were processed, based on the default Chl-*a* algorithm for MODIS (OC3), using NASA's SeaDAS (SeaWIFS Data Analysis System) software for the coastal region off Concepción. MODIS level 1 files were obtained from NASA (http://oceancolor.gsfc.nasa.gov/cms/) and processed using an improved atmospheric correction method for coastal turbid waters (Wang and Shi, 2007). Further details regarding processing





options and flags are found in Saldías et al. (2012). Composites of 3 days were generated to reduce the cloud cover and increase the number of pixels available during the sampling dates.

### 2.4 Satellite-derived wind stress and in-situ data

Satellite-derived, level 2 coastal ocean surface wind vectors were retrieved from the Advanced Scatterometer (ASCAT) on MetOp-A, at 12.5 km sampling resolution. Data files are obtained from the Royal Netherlands Meteorological Institute (http://www.knmi.nl/scatterometer/publications/pdf/ASCAT). For the study area, particular swaths presented wind measurement data from ~ 20-25 km from the coast, with improved wind vectors over the continental shelf around Sta. 18. In situ wind data were obtained from the Airport Carriel Sur meteorological station.

Daily means of wind stress were derived following Large and Pond (1981). Considering the synoptic character of the upwelling events, and to obtain the water mass residence time at the sampling station, box plots representing the boundaries of statistical values, and the error of in in-situ wind stress was calculated for the 5 days previous to sampling. To calculate the onset (i.e., spring transition) and extension of the upwelling season, cumulative alongshore (south-north) wind stress was estimated as described by Barth et al. (2007).

### 2.5 $^{15}$N tracer and inhibitor experiments

Experimental assays with $^{15}$N-labeled tracers and specific-pathway inhibitors were carried out with water from the oxycline and the bottom layer (25 m and 85 m depth, respectively), in order to evaluate the contribution and coupling of the different processes involved in N loss through the water column. The $^{15}$N-tracer incubations followed the basic procedures as previously described by Thamdrup et al. (2006), with some modifications. Basically, 250 mL glass bottles were filled immediately following water sampling with the Niskin bottles, to avoid oxygenation, allowing overflow until the volume was replaced three times. The bottles were closed with butyl rubber septa and crimps, and kept in the dark at a low temperature until later processing in the laboratory. $^{15}$N-labeled fixed N substrates were added through the septa into separate bottles as follows (the final concentration is stated in μmol L$^{-1}$ in parentheses): $^{15}$NO$_3^-$ (15), $^{15}$NO$_2^-$ (5), and $^{15}$NH$_4^+$ (5). In order to determine the $^{15}$N mole fraction, concentrations of the substrates were measured before and after addition of the $^{15}$N-labeled compounds. Treatments from the oxycline during January were evaluated under in situ O$_2$ conditions, while the rest of the samples were purged with helium during 15 min for anoxic incubations (both depths during September and the bottom samples in January). After adding the substrates, mixing, and purging (if necessary), water was dispensed into 12.6 mL glass vials (Exetainers, Labco) and incubated at in situ temperatures (10-11 °C) for up to 48 h. Subsamples were sacrificed in triplicate at four instances during the incubation (0, 15, 24, 48 h), and biological activity was halted by injection of 50 μL of a 50 % (w/v) zinc chloride (ZnCl$_2$) solution into each Exetainer. The N-isotopic composition of $^{15}$N$_2$ produced by anammox and/or denitrification was measured by gas chromatography continuous-flow isotope ratio mass spectrometry (GC-CF-IRMS; Finnigan Delta Plus) using N$_2$ as standard (Dalsgaard et al., 2012).




Subsequently, $^{15}N$ in $N_2O$ were analyzed from the same samples following the chromatographic separation of $N_2O$ on a GC column.

Specific inhibitors were used to block the activity of target pathways/microorganisms and discriminate the contribution of the different processes to N cycling and $N_2$ and $N_2O$ production. Allylthiourea - ATU (at a final concentration of 86 µmol $L^{-1}$), a specific bacterial inhibitor of the ammonium monooxygenase (AMO) enzyme (Ginestet et al., 1998), and N1-Guanyl-1, 7 Diaminoheptane - GC7 (at a final concentration of 100 µmol $L^{-1}$) a phylogenetic inhibitor of the archaeal growth (Jansson et al., 2000), were used in $^{15}NH_4^+$ incubations to separate the bacterial and archaeal contribution to ammonium oxidation (AO), and to indicate any possible competition for $^{15}NH_4^+$ between ammonium oxidizers and anammox during both sampling periods. Acetylene (final concentration of 10 % v/v), that inhibits the AO and the final step of denitrification (i.e., $N_2O$ to $N_2$; Balderston et al., 1976), was added to the $^{15}NO_2^-$ treatments during January to quantify the $N_2O$ produced from nitrite reduction (NiR). This experimental design is detailed in Table 1.

### 2.6 $^{15}N$-labeled dissolved inorganic N species

Pathways indirectly coupled to N loss processes were measured in the same incubations from dissolved inorganic N compounds formed by redox reactions (i.e., $^{15}NH_4^+$ from $^{15}NO_x^-$ or $^{15}NO_2^-$ from $^{15}NH_4^+$ and $^{15}NO_3^-$). After analyzing $^{15}N_2$ and $^{15}N_2O$ in the headspace of the Exetainers, the dissolved $^{15}N$-labeled species ($^{15}NH_4^+$ and $^{15}NO_2^-$) remaining in the liquid phase from the different treatments, were transformed into $^{15}N_2$ and analyzed by GC-CF-IRMS as previously described. To determine DNRA rates, $^{15}NH_4^+$ accumulated in $^{15}NO_x^-$ treatments was converted to $^{15}N$-$N_2$ using an alkaline hypobromite solution (Warembourg, 1993). Furthermore, rates of AO and nitrate reduction (NaR) were calculated as the mole fraction of $^{15}NO_2^-$ produced in $^{15}NH_4^+$ and $^{15}NO_3^-$ treatments, respectively, after conversion of the $^{15}NO_2^-$ produced by these pathways to $^{15}N$-$N_2$ with sulfamic acid (McIlvin and Altabet, 2005).

### 2.7 Natural abundance of nitrate isotopes

The natural abundance isotopic composition of N in $NO_3^-$ was analyzed by the denitrifier method (Casciotti et al., 2002, Sigman et al., 2001), after removal of $NO_2^-$ present in the samples by treatment with sulfamic acid (McIlvin and Altabet, 2005). The $\delta^{15}N$ values were measured in the $N_2O$ quantitatively produced after nitrate reduction by denitrifying bacteria that lack $N_2O$-reductase activity. The isotopic composition was analyzed in a GC-CF-IRMS (Finnigan Delta Plus). Data are reported in delta notation as $\delta^{15}N = (R_{(sample)}/R_{(reference)} - 1) \times 1000)$, $R = {^{15}N}/{^{14}N}$.

### 2.8 Calculations

$N_2$ produced by anammox and denitrification, and from the chemical conversion of $NO_2^-$ and $NH_4^+$ to $N_2$, was determined from the excess of $^{15}N$-labeled $N_2$ over the $^{15}N$ mole fraction in the source compounds according to Thamdrup et al. (2006), taking into account the random and hybrid isotope pairing patterns associated with



denitrification and anammox, respectively. Rates were derived from the slope of the linear regression of $^{15}$N-$N_2$ production ($^{14}N^{15}N$ and $^{15}N^{15}N$) as a function of incubation time. Only significant ($p \leq 0.05$) and linearly developing rates without lag phase were considered. Rates of both $NO_2^-$ and $NH_4^+$ production were calculated from linear regression of the concentration of the $^{15}$N-labeled product over time. Rates of $N_2O$ production are reported as separate rates of $^{45}N_2O$ ($^{14}N^{15}NO$) and $^{46}N_2O$ ($^{15}N^{15}NO$) accumulation, also from linear regression, as different pathways with different patterns of isotope pairing may contribute to $N_2O$ formation.

N deficits (N*) were calculated as stoichiometric anomalies from the Redfield ratio using the relation between fixed inorganic N and phosphate concentrations (Gruber and Sarmiento, 1997).

## 3 Results

### 3.1 Oceanographic setting

The hydrographic variability over the continental shelf off Concepción showed the steepening of the $O_2$ gradients as well as other chemical species, during the development of the austral upwelling season (September to April). The onset of the upwelling season during spring (early September), was characterized by a clear shift in the wind direction to predominantly southerly winds (arrows in Fig 1a and positive wind stress values denoted as red bars in Fig 1b), and the wind-driven incursion (up to 25 m depth) of cold, saline, $NO_3^-$-rich and $O_2$-poor ESSW (identified by the 11°C isotherm and 34.6 isohaline) (Fig 2a and 2b). This onset was also indicated by a change to positive values in the cumulative alongshore wind stress, that initiates on the 9$^{th}$ September 2013 (data not shown). This injection of nutrients, combined with a high availability of photosynthetically active radiation (PAR) triggered the progressive enhancement of near surface primary production (see the temporal increases in satellite-derived Chl-*a* in Fig 1a). This situation is intensified during summer, with stronger southerly wind pulses (Fig. 1a) and further shoaling of the ESSW to 20 m (Fig 2d). During this period also the exhaustion of $O_2$ intensified at bottom (< 5 µmol $L^{-1}$). Thus, three layers are established in the water column as a consequence of the influx of ESSW (Farías et al., 2009): a well oxygenated mixed layer, the oxycline characterized by a marked reduction of the dissolved $O_2$ content over a compressed vertical scale, and the bottom layer, where the oxygen deficiencies led to potentially anoxic conditions (Murillo et al., 2014). Furthermore, an increased availability of $NO_3^-$ and $NO_2^-$ as electron acceptors, and of organic matter, $NO_2^-$, $NH_4^+$, and $H_2S$ as donors, are also characterized by the intensification of upwelling through the two deeper layers (Galán et al., 2014).

During September, $NO_3^-$-rich waters (up to 30 µmol $L^{-1}$) occupied most of the subsurface, from the bottom to the upper boundary of the oxycline (up to 25 m depth; Fig. 2b). Nevertheless, although similar high $NO_3^-$ values remained through the oxycline during summer (January; Fig. 2e), notable $NO_3^-$ depletion (values as low as 19.5 µmol $L^{-1}$) was observed close to the bottom and associated with extreme $O_2$ exhaustion (~ 4 µmol $O_2$ $L^{-1}$), with a subsequent $NH_4^+$ and $NO_2^-$ accumulation (up to 1.3 and 1.6 µmol $L^{-1}$, respectively). N





deficiencies (N*) generally increased with depth (Fig 2c and 2f), with the greatest depletion (-24 µmol L$^{-1}$) observed in bottom waters during summer, which seems to be associated with the NO$_3^-$ consumption; while similar N* values were observed in the oxycline during both periods ($\sim$ -10 µmol L$^{-1}$). The vertical distribution of N$_2$O was bimodal with similar peak values at the oxycline and in bottom waters, with concentrations slightly increasing from spring to summer ($\sim$ 25 to 27 nmol L$^{-1}$, respectively; Fig. 2b and 2e).

During the spring transition (September), the maximum concentration and mean particle diameter were observed in the surface layer (120 µL L$^{-1}$ and up to 77 µm, respectively), possibly due to the building up of phytoplanktonic biomass, with a secondary peak occurring close to the base of the oxycline (36 µL L$^{-1}$ and 47 µm, respectively; $\sim$ 25 m depth). During summer, the particle concentration decreased relative to the spring, but showed a noteworthy increase in size (an entire order of magnitude greater than in September). During this period, particles exhibited a broad distribution along the oxycline with a maximum size and concentration at the upper boundary (up to 650 µm and 20 µL L$^{-1}$, respectively; $\sim$ 10 m depth), and a subsurface secondary peak ($\sim$ 500 µm and 11 µL L$^{-1}$, respectively; $\sim$ 25 m depth) around the base of this layer (Fig 2c and 2f).

### 3.2 Natural abundance of nitrate isotopes

The vertical distribution of $\delta^{15}$N-NO$_3^-$ showed contrasting patterns between the two sampling periods (Fig 2c and 2f). During spring, the $\delta^{15}$N value was highest at the surface (14.6 ± 0.3 ‰) and decreased consistently through the water column to a minimum at the bottom (11.2 ± 0.2 ‰). In summer, the $\delta^{15}$N values increased from surface to depth, ranging from 9.5 ± 0.3 ‰ to 14.5 ± 0.1 ‰, respectively. The oxycline showed relatively similar $\delta^{15}$N values for both periods, being slightly higher during spring (around 12 ‰ and 11 ‰, respectively).

### 3.3 $^{15}$N experiments

The production of labeled N-gaseous species (measured as N$_2$ and N$_2$O) revealed the coexistence of nitrification, anammox, and denitrification in both the oxycline and bottom layer during the study period (Table 2 and Figs. 3 and 4), with a remarkable increase in the N removal between spring and summer. Interestingly, during both seasons a higher activity in N loss was observed in the oxycline relative to the suboxic bottom waters, where activity was < 40 % of the oxycline values.

#### 3.3.1 $^{15}$N$_2$ production

During the spring transition (September; Figs. 3a, b), denitrification was the main N removal pathway in the oxycline (18.9 ± 4.3 nmol L$^{-1}$ d$^{-1}$) in comparison to the activity of anammox (2.8 ± 0.3 nmol L$^{-1}$ d$^{-1}$), with rates corresponding to 87 % and 13 % of the total N$_2$ produced in this layer, respectively. On the contrary, anammox was the only process involved in N$_2$ production in bottom waters (4.5 ± 0.9 nmol L$^{-1}$ d$^{-1}$). During summer (January; Figs. 3c, d), a similar vertical segregation in N$_2$ production processes was observed, however the magnitude of the rates changed. While at the oxycline, denitrification registered the highest N$_2$





production rate during this study ($50.3 \pm 10.3$ nmol L$^{-1}$ d$^{-1}$ - 73 % of total N$_2$ production at this layer), anammox' contribution to N$_2$ production in the oxycline increased substantially ($18.7 \pm 4.6$ nmol L$^{-1}$ d$^{-1}$ - 27 % of total N$_2$ production at this layer) relative to that observed during the spring transition. At depth, N$_2$ was only formed from anammox but at a very low rate ($0.4 \pm 0.02$ nmol N$_2$ L$^{-1}$ d$^{-1}$), that represents a substantial

reduction of this process there in comparison to the spring transition.

During summer, anammox activity increased following the addition of the AO inhibitor ATU (Fig. 3c). During this period, anammox rates from $^{15}NH_4^+$ treatments at the oxycline increased from $17.6 \pm 1.7$ nmol L$^{-1}$ d$^{-1}$ to $26.1 \pm 4.6$ nmol L$^{-1}$ d$^{-1}$, in incubations amended with ATU. Similar trends were observed at depth, but

with a relatively lower increase (from $0.4 \pm 0.02$ nmol L$^{-1}$ d$^{-1}$ from $^{15}NH_4^+$ to $0.5 \pm 0.04$ nmol L$^{-1}$ d$^{-1}$ from $^{15}NH_4^+$ + ATU). By contrast, no significant increase was observed after addition of GC7. At depth, anammox activity was not detected after the addition of this archaeal inhibitor.

### 3.3.2 $^{15}N_2O$ production

During the investigation, N$_2$O production from $^{15}N$ labeled substrates showed vertical and temporal changes

with respect to substrate source (Table 2 and Figs. 4a, d). In both sampling periods in the oxycline, N$_2$O was produced exclusively from AO, with slightly higher rates during the spring transition ($3.8 \pm 0.7$ nmol L$^{-1}$ d$^{-1}$), relative to the summer ($2.2 \pm 0.01$ nmol L$^{-1}$ d$^{-1}$). At depth, there was a change in the N$_2$O source pathway over time, however it should be noted that acetylene (C$_2$H$_2$) was not tested in the spring incubations. In September, N$_2$O was only produced from ammonium ($5.8 \pm 0.6$ nmol L$^{-1}$ d$^{-1}$), whereas production from ammonium was

lower ($0.02 \pm 0.002$ nmol L$^{-1}$ d$^{-1}$) relative to the N$_2$O produced from nitrite ($0.05 \pm 0.005$ nmol L$^{-1}$ d$^{-1}$) during January. Nevertheless, the addition of C$_2$H$_2$, which inhibits N$_2$O reduction to N$_2$, resulted in a higher production of N$_2$O (up to $22.6 \pm 3.5$ nmol L$^{-1}$ d$^{-1}$ from $^{15}NO_2^-$ + C$_2$H$_2$). This N$_2$O accumulation was not observed in the oxycline during summer, despite C$_2$H$_2$ also being added into the NO$_2^-$ incubations.

### 3.3.3 Dissolved inorganic $^{15}N$-compounds cycling

The redox activity related to the production of NO$_2^-$ and NH$_4^+$, substrates of denitrification and anammox, is summarized in Figure 5 and Tables 3 and 4, respectively. In general, nitrite production was higher during summer (Fig. 5b) than in the spring transition (Fig. 5a), but with variation observed in maximum production activity between layers. During September, total NO$_2^-$ produced (i.e., NO$_2^-$ formed from AO plus NaR) was higher in the bottom waters (15.1 nmol NO$_2^-$ L$^{-1}$ d$^{-1}$) relative to the oxycline (8.0 nmol NO$_2^-$ L$^{-1}$ d$^{-1}$), with AO

and NaR contributing similarly in both layers. On the contrary, in January nitrite production was higher in the oxycline with NaR as the main source ($83 \pm 5$ nmol NO$_2^-$ L$^{-1}$ d$^{-1}$ vs. $53 \pm 16$ nmol NO$_2^-$ L$^{-1}$ d$^{-1}$ from NaR and AO – 61 and 39 %, respectively), while AO was the principal source of NO$_2^-$ at depth ($38 \pm 6$ nmol NO$_2^-$ L$^{-1}$ d$^{-1}$ vs. $16 \pm 4$ nmol NO$_2^-$ L$^{-1}$ d$^{-1}$ from AO and NaR – 70 and 30 %, respectively).





During summer, the addition of ATU into the incubations resulted in a significant reduction (> 70 %) in nitrite production from AO, both in the oxycline (14.4 ± 4.9 nmol $NO_2^-$ $L^{-1}$ $d^{-1}$) and at depth (9.6 ± 0.7 nmol $NO_2^-$ $L^{-1}$ $d^{-1}$), while GC7 caused only a slight but non-significant reduction in nitrite production rates from AO at both depths. By contrast, during the onset of the upwelling season, no nitrite production activity was observed from AO for treatments with either ATU or GC7 addition, indicating that AO could be completely inhibited by these compounds (data not included in Table 3).

Similar to the nitrite production scenario, over both depths the labeled $NH_4^+$ formed by dissimilative reduction of $NO_2^-$ (i.e., DNiRA) was higher during summer (Fig. 5d) relative to the spring transition (Fig. 5c), with marked variations in the yield of this reaction between depths throughout the upwelling season. In September, DNiRA was similarly active in the oxycline (59.9 ± 3.9 nmol $NH_4^+$ $L^{-1}$ $d^{-1}$) and in the bottom layer (49.3 ± 5.3 nmol $NH_4^+$ $L^{-1}$ $d^{-1}$). On the contrary, during January a notable production of $NH_4^+$ was observed in the oxycline (up to 751 ± 81 nmol $NH_4^+$ $L^{-1}$ $d^{-1}$) in comparison to the production at depth (82.8 ± 7.3 nmol $NH_4^+$ $L^{-1}$ $d^{-1}$).

## 4 Discussion

### 4.1 Environmental variability

The hydrographic properties during the study period reflects the already well-described onset of the upwelling season during the spring transition (September), and the subsequent development and intensification of upwelling conditions in the summer as upwelling progresses over the shelf off central Chile (Farías et al., 2009, Galán et al., 2012). Southerly wind pulses during spring-summer (Fig. 1b) draw ESSW with low $O_2$ (< 50 µmol $L^{-1}$) and a high amount of $NO_3^-$ (> 25 µmol $L^{-1}$) onto the shelf from the southernmost extension of the permanent ESP OMZ (Galán et al., 2014). As the upwelling season progresses, heterotrophic processes in subsurface waters are stimulated by the large amount of organic matter produced at the surface that settles on the sediments. The increase in respiration rates creates extreme $O_2$ depletion at depth during the summer (∼ 4 µmol $L^{-1}$; Fig. 2d) and occasionally anoxic conditions occur in the bottom waters of this system (Murillo et al., 2014). The $O_2$ limitation switches on anaerobic microbial metabolisms, which initially use $NO_3^-$ as the terminal electron acceptor generating other inorganic species of N (i.e., $NH_4^+$, $NO_2^-$, $N_2O$; Fig. 2e), which in turn may support chemolitho- and chemoorgano-trophic metabolisms including nitrification, anammox, and denitrification, contributing in this way to the loss of fixed N.

The coupling between primary production at the surface and anaerobic N-based metabolisms in the subsurface depends on alternations between upwelling-favorable wind pulses and relaxed or inverted wind events, which allows for organic particles to settle on the sediment. In this regard, clear variations in the alongshore wind stress (intensity and direction) were observed between sampling times, considering the 5 days preceding the sampling and the sampling day itself. Thus, the system was coming from a pulse of





inverted, downwelling-favorable winds during the sampling of the spring transition, in contrast to a transitional upwelling-favorable situation during the sampling in summer (see box plots in Fig. 1b). These variations in wind characteristics also determine the intensity of near-surface primary production, illustrated by the Chl-*a* distribution as a proxy of organic matter formation (Fig. 1a), and the availability, size, and

vertical distribution of particles through the water column (Figs. 2c, f). Hence, during September a less productive system was observed, with a secondary peak of small particles more narrowly distributed around the oxycline base, whereas in January a more intense vertical advection of ESSW supported an increase in surface Chl-*a*, with a wider distribution of larger particles around both oxycline limits.

**4.2 $^{15}$N-loss activity**

An intense N cycling leading to N-loss over the continental shelf off central Chile was demonstrated by the $^{15}$N labeling results (Figs. 3, 4, and 5), with higher levels of activity observed not only as the upwelling season progressed, but also in the oxycline (relative to the bottom waters), during both periods. This is the first time that anammox, denitrification, and nitrification – through $N_2O$ production, were found co-contributing to the

removal of fixed N in this coastal system, and at relatively higher $O_2$ conditions as measured in the oxycline ($\sim$ 95 $\mu$mol $O_2$ $L^{-1}$). A plausible explanation for the observation of oxygen-sensitive processes (i.e., anammox and denitrification) operating under ambient oxic conditions is that the oxycline incubations for both sampling periods were carried out with water retrieved from the secondary peak of particles ($\sim$ 25 m depth; Figs. 2c, f), which accumulate at this level due high stratification (Figs. 2a, d), causing a deceleration in their downward

flux. These organic particles may create anoxic microenvironments (Ploug et al., 1997), as well as generate chemical hotspots that provide substrates (organic and inorganic) to sustain the anaerobic metabolisms of the pelagic microbiota. The influence of anaerobic microniches on $N_2$ production has been previously proposed, in scenarios where strong covariation between anammox rates and the abundance of particle-associated anammox cells have been found in incubations with water from the Namibian continental shelf, with in situ

$O_2$ levels reaching 25 $\mu$mol $L^{-1}$ (Kuypers et al., 2005, Woebken et al., 2007). Recently, several anaerobic N processes (e.g., nitrate reduction, denitrification, anammox, DNRA, and $N_2O$ production) have been detected in suspended cyanobacterial and diatom anoxic-aggregates (traced with $O_2$ microsensors) incubated in oxic waters (Klawonn et al., 2015 and Stief et al., 2016, respectively), supporting the idea that particles are able to operate as an anoxic N-removal hotspot. Particles in this study were smaller ($\leq$ 0.6 mm) than the

cyanobacterial/algal aggregates (diameters $\geq$ 3 mm; Klawonn et al., 2015 and Stief et al., 2016, respectively), but the static incubation may have facilitated the development of anoxic niches in our incubations by allowing the particles to settle. Activity of anoxic metabolism in these $O_2$-depleted microenvironments is related to the size of the particles/aggregates and the ambient oxygen conditions, due to the availability of substrate and the diffusion velocity, respectively. Larger particles in situations of low ambient $O_2$ levels offer more organic

substrates, that favor internal $O_2$ consumption through respiratory activity, facilitating the development of anoxia within the particles core (Ploug et al., 1997; Ploug, 2008; Klawonn et al., 2015). Thus, differences in



particle size could explain differences in the N loss activity observed at the oxycline between sampling periods.

Phylogenetic analyses of functional genes indicate that this coastal system maintains a great diversity of pelagic N cycling microorganisms, with heterotrophic denitrifying bacteria, aerobic ammonia oxidizers (either from bacteria and archaea domains), and anammox bacteria related with the *Candidatus* Scalindua clade, present not only in the oxygen-depleted bottom waters but also around the oxycline (O Ulloa unpublished data). Furthermore, previous evidence of the presence of anammox cells in the study area, identified by CARD-FISH, showed that this bacterial population could be distributed throughout the water column, at similar higher numbers, when the maximal development of upwelling is reached (Galán et al., 2012).

The presence of $O_2$ (in situ) in particle-associated incubations from the oxycline during summer, favors coupling between aerobic mechanisms (e.g., AO and organic matter remineralization) with anaerobic N removal processes (i.e., anammox and denitrification), as illustrated by the high production rates of $NO_2^-$ and $NH_4^+$ observed in January (Fig. 5). Assuming that these aerobic processes also occur in association with the particles, it is proposed that aerobic metabolisms occurring at the periphery of the oxygenated particle fuel the internal anaerobic pathways by the diffusion of their products (e.g., nitrite produced by AO) to the respiration-mediated anoxic core, similar to the scenarios described for various other microbial aggregates in wastewater treatment plants (Nielsen et al., 2005), manmade model systems (Stief et al., 2016) and natural environments (Klawonn et al., 2015).

In summer, $N_2$ production by heterotrophic denitrifier communities in the oxycline was higher than during the spring transition. This can be mostly accounted by differences in the availability of reactive organic matter, consistent not only with the rise in Chl-*a* (Fig. 1a), but also with the increase in the observed particle size (Fig. 2c, f). The influence of organic matter on denitrification activity has been extensively documented in OMZ waters (e.g., Ward et al., 2008; Ward et al., 2009; Dalsgaard et al., 2012; Chang et al., 2014). Furthermore, the increased nitrite production rates observed in the oxycline during summer (Figs. 5a, b), and the in situ accumulation of this substrate (see profile in Fig. 2e), could also favor denitrification; although the main electron acceptor for this process was $NO_3^-$, which is abundantly available, as demonstrated by the differences in $N_2$ production from $^{15}NO_2^-$ and $^{15}NO_3^-$ amended treatments (Figs. 3b, d). Similar to denitrification, anammox rates in the oxycline also increased from spring to summer. This temporal variability in anammox activity could be related with the availability of $NO_2^-$ and $NH_4^+$, which is implied by the observation of a vast increase in the production of these substrates during January (Figs. 5b, d). The intense cycling of $NO_2^-$ and $NH_4^+$ via nitrification, anammox, and denitrification, among other processes (e.g., nitrite and ammonium assimilation; processes not determined in this study), should prevent a substantial in situ accumulation of these substrates (Fig. 2e).



At depth, $N_2$ production rates by anammox and denitrification were either reduced or non-existent in comparison to the rate of activity observed in the oxycline. Furthermore, rates were lower that those previously observed in the same area (Galán et al., 2014). This reduction in $N_2$ production throughout the water column, and over time (intra- and inter-seasonally) is likely to be related with variations in the

availability of $O_2$ and substrates that fuel both processes (i.e., organic matter, $NO_2^-$, and $NH_4^+$). A greater depletion of ambient $O_2$ could explain the major N removal activity that was previously observed in the area (Galán et al., 2014), a scenario that has been widely observed elsewhere (Jensen et al., 2008; Kalvelage et al., 2011; Dalsgaard et al., 2012). On the other hand, vertical variations in the quality of the organic matter that sinks to the depths could affect microbial organoheterotrophic activity, including denitrification rates (Chang

et al., 2014). In turn, this affects the delivery of remineralized compounds required for lithoautotrophic processes, such as anammox and nitrification, among others. Likewise, reduced anammox and organotrophic denitrification activity has been previously registered during the summer period at depth, and was related to the presence of reduced sulfur compounds in the water column that inhibited the activity of the anammox cells (Jensen et al., 2009), and triggered a lithoautotrophic, $H_2S$-based metabolism of the denitrifier

community (Galán et al., 2014).

It can be concluded that in this coastal system during summer the main substrate source for $N_2O$ production changes along chemical gradients through the water column. Thus, $N_2O$ was produced from $NH_4^+$ within the oxycline, while at depth it was produced from $NO_2^-$. However, it must be considered that during the spring

transition, $C_2H_2$, a $N_2O$ reduction inhibitor after NiR, was not added to the incubations. Conversely, while the NiR pathway is associated with the production of $N_2O$, based on the experimental design used here, it is not clear which pathway formed $N_2O$ coupled to the oxidation of $NH_4^+$. It is possible that $N_2O$ was produced by bacteria during the oxidation of $NH_4^+$ to $NO_2^-$ as a side product of the reaction (e.g., Goreau et al., 1980), or through the less understood nitrifier-denitrification reaction (e.g., Wrage et al., 2001). The latter pathway

combines two labeled N atoms from NO or $NO_2^-$, and should therefore, if these substrates originate from AO, produce mostly $^{46}N_2O$ in our experiment, whereas $^{45}N_2O$ dominated (Table 2 and Fig. 4). Ammonium oxidizing archaea may also participate in $N_2O$ production (Martens-Habbena et al., 2009; Santoro et al., 2010; Santoro et al., 2011; Jung et al., 2014). In this sense, the large dominance of $^{45}N_2O$ over $^{46}N_2O$ observed, could be explained by the recently described "hybrid" pathway (Kozlowski et al., 2016), where labeled

hydroxylamine ($^{15}NH_2OH$) produced from AO might be combine with NO reduced from the $NO_2^-$ present in the native pool of the incubations. The coupling between AO and NiR, after the formation of $NO_2^-$, is less plausible because any labeled $NO_2^-$ produced would be diluted by the existing pool of $NO_2^-$ in the environment. Furthermore, experiments carried out with $^{15}NO_2^-$ in the oxycline during summer did not produce labeled $N_2O$ via NiR, despite the use of $C_2H_2$ as was observed at depth, rejecting the idea that the

coupling between AO and NiR is the pathway through $N_2O$ is generated. Increases in $N_2O$ production within the oxycline have also previously been associated with AO (Codispoti, 2010).





It appears more likely that differences in $N_2O$ produced in the oxycline between seasons are related to $O_2$ concentration, rather than to $NH_4^+$ availability, as the production of this substrate was consistently at least of one order of magnitude higher than the ammonium-based $N_2O$ production rates (Figs. 4c, d). The $O_2$-limited incubations carried out during September, relative to in situ $O_2$ conditions during the January experiments,

could, together with a greater availability of particles occurring at the sampling depth, generate an $O_2$-depleted environment, in which the yield of $N_2O$ formed through nitrifier-denitrification (Elkins et al., 1978; Goreau et al., 1980; Lipschultz et al., 1981; de Wilde and de Bie, 2000) or through the "hybrid" pathway (Kozlowski et al., 2016) will increase. There is ample evidence that $N_2O$ production via nitrification in aquatic systems is increased under conditions of reduced oxygen and abundant ammonium availability (Elkins

et al., 1978; McElroy et al., 1978; de Wilde and de Bie, 2000).

During summer, at least for the duration of the sampling period of this study bacteria seemed to play a major role for the modulation of AO compared to their archaeal counterpart based on the different effects of ATU and GC7 inhibitors on nitrite production (Fig. 5). This result could also indicate an in situ influence of

ammonium oxidizers on the N loss activity through competition for $NH_4^+$ with anammox cells (Fig. 3), and also possibly through the involvement in $N_2O$ production. This segregation between AO players was not observed during the spring transition, probably owing to differences in $O_2$ availability during the incubations that could suppress the activity of ammonium oxidizers as previously mentioned. However, a dominance of bacterial AO does not correspond with recent molecular and biogeochemical observations from the study

area, particularly during the upwelling season, when high abundance of *Thaumarchaeota* (previously *Crenarchaeota*; an archaeon that sustains itself chemolithoautotrophically through aerobic oxidization of $NH_4^+$) account for a significant fraction of the microbial assemblage, while ammonium-oxidizing bacteria are scarce or undetectable (Levipan et al., 2007; Quiñones et al., 2009; Bristow et al. 2016). Likewise, the relative contribution to ammonium oxidation, quantified by the ammonia monooxygenase subunit A gene (Molina et

al., 2010), and dark carbon assimilation rates (Farías et al., 2009), showed that archaea had a greater potential activity relative to the homologous bacteria in the system. Also, the *Thaumarchaeota* are mainly associated with the oxycline (Belmar in preparation), where $N_2O$ hotspots are frequently observed (Cornejo and Farías, 2012), further suggesting their possible participation in $N_2O$ production in this layer, particularly as the isotopic signature suggests ammonium oxidation by archaea is a major precursor of $N_2O$ in the ocean

(Santoro et al., 2011). To reconcile the inhibitor results with all other evidence, we speculate that GC7, as a growth inhibitor of archaea, might not affect archaeal metabolism in short-term incubations, thus masking the relative importance of this domain for AO during this study.

During the sampling period $NH_4^+$ was produced at a high rate by dissimilative nitrite reduction (i.e., DNiRA);

a process measured for the first time in this system. Rates were especially high during summer in the oxycline. It should be noted, however, that the hypobromite method used to analyze $^{15}NH_4^+$ also oxidizes organic $^{15}N$, which may have formed through assimilation of labeled $NO_2^-$. Nonetheless, it is remarkable that all the $NH_4^+$ required for the lithoautotrophic community, i.e. anammox and nitrification (measured here as





AO), and even for previous rates of light assimilation measured at the same station (spring-summer season average < 100 and < 50 nmol L$^{-1}$ d$^{-1}$ for 30 and 80 m, respectively; Fernandez and Farías, 2012), could be supported by DNiRA at the two depths evaluated. Furthermore, the expected ammonium production from organic matter remineralization by canonical denitrification and nitrate and nitrite reduction, at the rates

measured here and assuming that these are heterotrophic processes, should be an important additional source of this substrate. The DNRA process was suggested to be important in some previous surveys in the OMZ off Peru and in the Arabian Sea (Lam et al., 2009; Jensen et al., 2011; but see also Kalvelage et al., 2013) and was suggested to supply anammox with ammonium in previous OMZ studies where heterotrophic denitrification was not detected (Thamdrup et al., 2006; Hamersley et al., 2007; Galán et al., 2009; Lam et al.,

2009). Nevertheless, the NO$_2^-$ required just to support the DNiRA, disregarding the NO$_2^-$ demand from other processes (e.g., anammox and denitrifiers, or for nitrite assimilation and nitrite oxidation - not evaluated in this investigation), was not fully supported by the measured nitrite production through nitrate reduction and ammonium oxidation. This uncoupling between microbial sources and sinks for this substrate, indicates that NO$_2^-$ should be a limiting factor and hence a pivotal compound in this system. Thus, allochthonous NO$_2^-$ is

needed to cope with this imbalance. Considering this, upwelled NO$_2^-$ from the oceanic OMZ is the most probably source (e.g., Galán et al., 2009), with an extensive vertical distribution starting close to the upper OMZ boundary and reaching concentrations up to 10 µmol L$^{-1}$. Alternatively, fluxes from sediments are a less probable source due the increased consumption of nitrite in this environment (Devol, 2015).

### 4.3 Natural abundance of nitrate isotopes

The enriched δ$^{15}$N-NO$_3^-$ values observed (exceeding the average in the deep Pacific of 5 ‰; Sigman et al., 2000) indicate that nitrate-based biological mechanisms are the main modulators of the isotopic N composition in the subsurface waters over the shelf off central Chile during spring and summer. The general negative trend between NO$_3^-$ concentration and δ$^{15}$N provides evidence that light isotopes are preferentially used by biologically mediated NO$_3^-$ consuming processes, leaving the residual NO$_3^-$ pool enriched. Likewise,

concurrence between the δ$^{15}$N values and the N deficit (except for the surface sample) confirms previous reports (Voss et al., 2001; Sigman et al., 2000). The δ$^{15}$N value (11.2 ‰) at depth during the spring transition is used as an estimate of the background source signal during the onset of the upwelling season. Thus, the progressive δ$^{15}$N enrichment observed throughout the oxycline to the surface may result from locally occurring NO$_3^-$ fractionation processes, including dissimilatory nitrate reduction processes and nitrate

assimilation. During summer, the association between the maximum δ$^{15}$N-NO$_3^-$ value (up to 14.5 ‰) at depth, the maximum N deficit (Figs. 2e, f), and a pronounced O$_2$ deficiency (< 5 µmol L$^{-1}$), is likely to be the result of diverse dissimilative nitrate consumption processes (i.e., nitrate reduction to nitrite, DNaRA, and denitrification through N$_2$O production). The isotope N ratio was reduced in the oxycline (11.7 ‰; Fig. 2f), possibly due the addition of isotopically light NO$_3^-$ by the oxidation of recently fixed N, with an isotopic

composition of ∼ -2-0 ‰ (Liu et al., 1996). Nitrogen fixation in this system occurs mainly during summer and fall (Fernandez et al., 2011), and the process may produce up to 20 % of the new N in surface waters



associated with OMZs (Brandes et al., 1998). In general, our observations are consistent with those reported previously for the Peruvian OMZ (Ryabenko et al., 2012), where high $\delta^{15}$N-NO$_3^-$ values (> 10 ‰) were associated with denitrification (and in this case together with nitrate reduction processes), occurring at lower O$_2$ concentrations (< 5 μmol L$^{-1}$), and higher N deficits. Furthermore, this is consistent with fractionation during nitrate assimilation by phytoplankton in surface waters (O$_2$ > 200 μmol L$^{-1}$).

## 5 Conclusions

Considering that N removal processes are initially fueled by organic matter and the concomitant production of electron donors (e.g., NH$_4^+$) and acceptors (e.g., NO$_2^-$) after its remineralization, the vertical and temporal differences in N-loss processes during this study, and previous studies in the area (Galán et al., 2014), highlight the influence of the wind pulses in the downward fluxes of organic matter particles. The wind pattern allows or prevents (when organic matter is exported off shore) that this material reaches the oxygen-limited layers. Thus, the wind regime shows a temporal frequency that is superimposed on the seasonal pattern, with the result that mechanisms of fixed-N removal, among other processes, vary intraseasonally in pulses ultimately controlled by the episodic vertical supply of organic matter. Furthermore, the accumulation of sinking organic particles around the oxycline considerably increases the volume for removing N processes in this coastal system. Nonetheless, and despite the great amount of regenerated ammonium produced after the oxidation of this organic load, and its central role in regulating the generation of N$_2$O at the oxycline, the main substrate that supports the N-removal as either N$_2$ and N$_2$O produced, is the overabundance of allochthonous N, mostly in the form of nitrate, which fertilizes the system through the upwelling season. Thus, during this productive period, pulses of accumulation and consumption of different N substrates modulated the structure and activity of the microbial N-based assemblage of this coastal upwelling system.

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



**Table 1**. Experimental design. Tracked N loss process and coupled pathways by the use of $^{15}$N-substrate, with detailed of the inhibitors added. Measured $^{15}$N products and phase are described.

| Tracer (concentration) | Process measured - Phase | Inhibitor (concentration) | Organism / Inhibited Process | Enzyme inhibited | Compound measured |
|---|---|---|---|---|---|
| $^{15}$NH$_4^+$ (5 µM) | Anammox and Denitrification - Gas | CONTROL | | | $^{15}$N$_2$ and $^{15}$N$_2$O |
| | Nitrification – Liquid* | | | | $^{15}$NO$_2^-$ |
| | Nitrification – Liquid* | ATU (90 µM) | Bacteria - Ammonium oxidation | Ammonia monooxygenase | $^{15}$NO$_2^-$ |
| | Nitrification – Liquid* | GC7 (100 µM) | Archaea - Ammonium oxidation | | $^{15}$NO$_2^-$ |
| $^{15}$NO$_2^-$ (5 µM) | Anammox and Denitrification - Gas | CONTROL | | | $^{15}$N$_2$ and $^{15}$N$_2$O |
| | DNRA – Liquid* | | | | $^{15}$NH$_4^+$ |
| | Denitrification - Gas | Acetylene (10% v/v) | Bacteria /Nitrite reduction | Nitrous oxide reductase | $^{15}$N$_2$O |
| $^{15}$NO$_3^-$ (15 µM) | Anammox and Denitrification - Gas | CONTROL | | | $^{15}$N$_2$ and $^{15}$N$_2$O |
| | Nitrate reduction – Liquid* | | | | $^{15}$NO$_2^-$ |

5    * Production of dissolved inorganic $^{15}$N compounds in the liquid phase was measured after chemical conversion of each substrate to $^{15}$N$_2$. For details see Sect. 2 Methods.



**Table 2**. Experimental dates, oxygen conditions during the incubations, tested depths, tracers and inhibitors amendments, measured N-gaseous specie (N$_2$ or N$_2$O), summary of the Student's t-test used to evaluate the significance of the activity obtained, and anammox (Amx) and denitrification (Dntf) rates.

| Sampling date | O$_2$ conditions | Depth [m] | Tracer | Inhibitor | Product measured | F-statistic | p-value | Significance | Degrees of freedom | Amx rate [nmol L$^{-1}$ d$^{-1}$] | Dntf rate [nmol L$^{-1}$ d$^{-1}$] | N$_2$O production [nmol L-1 d-1] |
|---|---|---|---|---|---|---|---|---|---|---|---|---|
| 12 September 2013 | Anoxic | 25 | $^{15}$NH$_4^+$ | WI | N$_2$ | 64,71 | 0,00400 | * | 4 | 2.214 ± 0.302 | | |
| | | | | | N$_2$O | 29,24 | 0,00567 | ** | 5 | | | 3.800 ± 0.700◆ |
| | | | $^{15}$NO$_2^-$ | WI | N$_2$ | 94,48 | 0,00063 | *** | 5 | 2.801 ± 0,300 | | |
| | | | | | N$_2$ | 52,41 | 0,00544 | ** | 4 | | 1.311 ± 0.201 | |
| | | | $^{15}$NO$_3^-$ | WI | N$_2$ | 21,37 | 0,00986 | ** | 5 | 1.523 ± 0.326 | | |
| | | | | | N$_2$ | 18,46 | 0,01268 | * | 5 | | 18.891 ± 4.339 | |
| | | 85 | $^{15}$NH$_4^+$ | WI | N$_2$ | 42,89 | 0,00281 | ** | 5 | 2.603 ± 0.400 | | |
| | | | | | N$_2$O | 108,83 | 0,00005 | *** | 7 | | | 5.800 ± 0.600◆ |
| | | | $^{15}$NO$_2^-$ | WI | N$_2$ | 23,97 | 0,00272 | ** | 7 | 4.524 ± 0.905 | | |
| 28 January 2014 | in situ | 25 | $^{15}$NH$_4^+$ | WI | N$_2$ | 100,09 | 0,00006 | *** | 7 | 17.622 ± 1.722 | | |
| | | | | ATU | N$_2$ | 32,19 | 0,01084 | * | 4 | 26.129 ± 4.557 | | |
| | | | | GC7 | N$_2$ | 56,46 | 0,00029 | *** | 7 | 19.952 ± 2.633 | | |
| | | | | WI | N$_2$O | 6.8x10$^{31}$ | 1.4x10$^{-32}$ | *** | 3 | | | 2.200 ± 0.000◆ |
| | | | $^{15}$NO$_2^-$ | WI | N$_2$ | 36,83 | 0,00175 | ** | 6 | 16.748 ± 2.774 | | |
| | | | | | N$_2$ | 11,54 | 0,02736 | * | 5 | | 5.547 ± 1.618 | |
| | | | $^{15}$NO$_3^-$ | WI | N$_2$ | 16,76 | 0,01492 | * | 5 | 18.726 ± 4.575 | | |
| | | | | | N$_2$ | 24,31 | 0,00787 | ** | 5 | | 50.370 ± 10.269 | |
| | Anoxic | 85 | $^{15}$NH$_4^+$ | WI | N$_2$ | 172,25 | 0,00576 | ** | 3 | 0.369 ± 0.025 | | |
| | | | | ATU | N$_2$ | 12,31 | 0,02470 | * | 5 | 0.523 ± 0.036 | | |
| | | | | WI | N$_2$O | 40,33 | 0,00315 | ** | 5 | | | 0.020 ± 0.002◆ |
| | | | $^{15}$NO$_2^-$ | WI | N$_2$O | 20,67 | 0,00613 | ** | 6 | | | 0.054 ± 0.005 |
| | | | | | N$_2$O | 86,36 | 0,00024 | *** | 6 | | | 0.007 ± 0.000 |
| | | | | C$_2$H$_2$ | N$_2$O | 40,43 | 0,00314 | ** | 5 | | | 22.630 ± 3.502 |
| | | | | | N$_2$O | 12,92 | 0,01144 | * | 7 | | | 3.872 ± 1.055 |
| | | | $^{15}$NO$_3^-$ | WI | N$_2$O | 56,25 | 0,00029 | *** | 7 | | | 0.023 ± 0.004 |
| | | | | | N$_2$O | 56,25 | 0,00029 | *** | 7 | | | 0.009 ± 0.001 |

Significance codes: 0.001 - ***, 0.01 - **, 0.05 - *, not significant – ns

Without inhibitor – WI

Degrees of freedom - DF

◆ - Number represent "partial rates" due it is not clear which pathway forms the N$_2$O from AO





**Table 3**. Experimental dates, oxygen conditions during the incubations, tested depths, tracers and inhibitors amendments, summary of the Student's t-test used to evaluate the significance of the activity obtained, and nitrite production rates measured as $^{29}N_2$ after chemical conversion.

| Sampling date | O₂ conditions | Depth [m] | Tracer | Inhibitor | F-statistic | p-value | Significance | Degrees of freedom | NO₂⁻ Production Rate ± SE [nmol L⁻¹ d⁻¹] |
|---|---|---|---|---|---|---|---|---|---|
| 12 September 2013 | Anoxic | 25 | $^{15}NH_4^+$ | WI | 42,51 | 0,00062 | *** | 7 | 3.649 ± 0.560 |
| | | | $^{15}NO_3^-$ | WI | 46,07 | 0,00106 | ** | 6 | 4.433 ± 0.653 |
| | | 85 | $^{15}NH_4^+$ | WI | 11,34 | 0,02809 | * | 5 | 7.830 ± 2.325 |
| | | | $^{15}NO_3^-$ | WI | 54,22 | 0,00181 | ** | 5 | 7.300 ± 0.991 |
| 28 January 2014 | *in situ* | 25 | $^{15}NH_4^+$ | WI | 10,81 | 0,03025 | * | 5 | 53.036 ± 16.127 |
| | | | | ATU | 8,50 | 0,03314 | * | 6 | 14.360 ± 4.927 |
| | | | | GC7 | 9,50 | 0,03516 | * | 6 | 46.141 ± 4.537 |
| | | | $^{15}NO_3^-$ | WI | 293,74 | 7x10⁻⁵ | *** | 5 | 82.918 ± 4.838 |
| | Anoxic | 85 | $^{15}NH_4^+$ | WI | 40,19 | 0,00072 | *** | 7 | 37.537 ± 5.921 |
| | | | | ATU | 92,53 | 4x10⁻⁵ | *** | 7 | 9.621 ± 0.749 |
| | | | | GC7 | 112,52 | 4x10⁻⁵ | *** | 7 | 32.321 ± 3.049 |
| | | | $^{15}NO_3^-$ | WI | 21,42 | 0,00982 | ** | 5 | 16.440 ± 3.552 |

Significance codes:  0.001 - ***, 0.01 - **, 0.05 - *, not significant – ns

Without inhibitor - WI





**Table 4**. Experimental dates, oxygen conditions during the incubations, tested depths, tracers amendments, pairing labeled $N_2$ product measured, summary of the Student's t-test used to evaluate the significance of the activity obtained, and ammonium production rates (DNRA) measured as $^{29/30}N_2$ after chemical conversion.

| Sampling date | $O_2$ conditions | Depth [m] | Tracer | Product measured | F-statistic | p-value | Significance | Degrees of freedom | $NH_4^+$ Production Rate ± SE [nmol $L^{-1}$ $d^{-1}$] |
|---|---|---|---|---|---|---|---|---|---|
| 12 September 2013 | Anoxic | 25 | $^{15}NO_2^-$ | $^{29}N_2$ | 154.26 | 0.00024 | *** | 5 | 55.979 ± 3.998 |
| | | | | $^{30}N_2$ | 43.51 | 0.00120 | ** | 6 | |
| | | 85 | $^{15}NO_2^-$ | $^{29}N_2$ | 12.74 | 0.02337 | * | 5 | 49.261 ± 5.278 |
| | | | | $^{30}N_2$ | 1095.96 | 0.00091 | *** | 3 | |
| 28 January 2014 | *in situ* | 25 | $^{15}NO_2^-$ | $^{29}N_2$ | 84.75 | 0.00077 | *** | 5 | 751.232 ± 81.459 |
| | | | | $^{30}N_2$ | 9.97 | 0.03423 | * | 5 | |
| | Anoxic | 85 | $^{15}NO_2^-$ | $^{29}N_2$ | 120.56 | 0.00039 | *** | 5 | 82.848 ± 7.310 |
| | | | | $^{30}N_2$ | 19.67 | 0.01137 | * | 5 | |

Significance codes:  0.001 - ***, 0.01 - **, 0.05 - *, not significant – ns





(a)

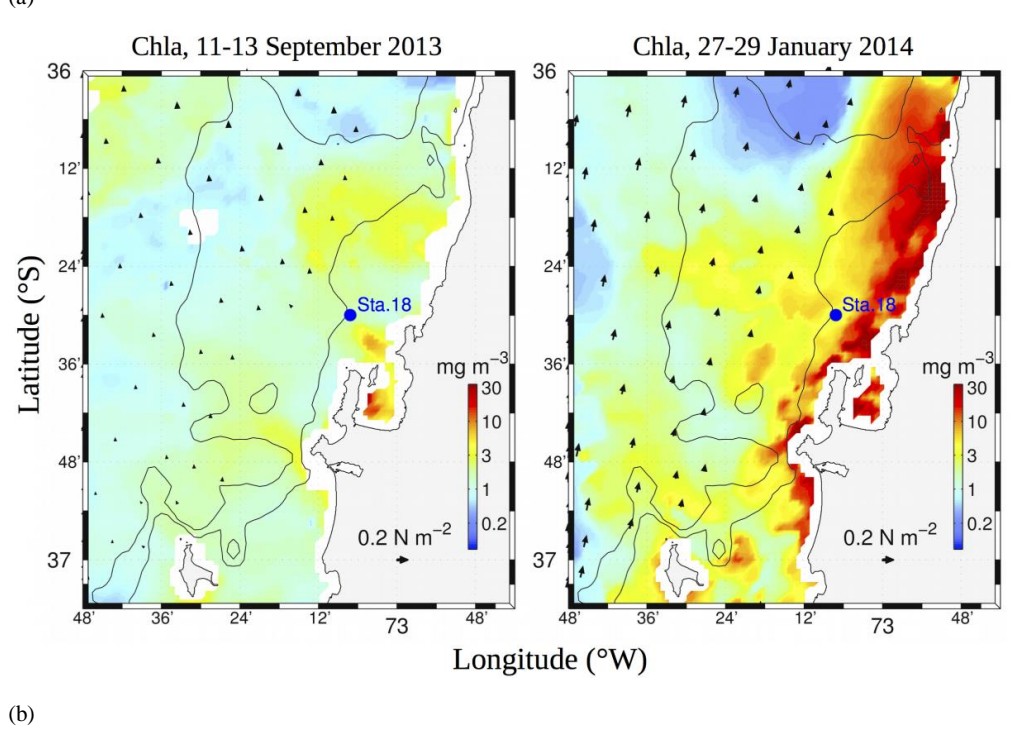

(b)

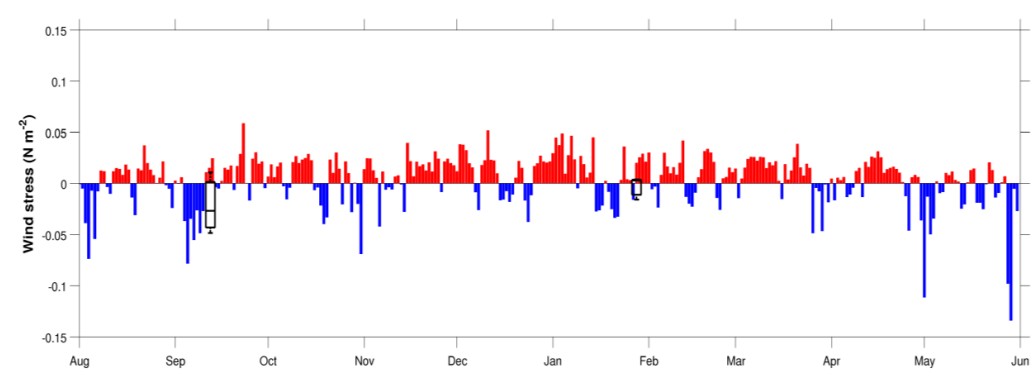

**Fig. 1**. (a) Satellite-derived chlorophyll-*a* and wind stress during the period around the sampling days (left panel spring transition and right panel summer), for the study area and location of time series Sta. 18 (blue dot; 36°30' S; 73°08' W) off central Chile, Concepción Bay. (b) Upwelling (red bars) and downwelling-favorable (blue bars) alongshore wind stress related to the study period (August 2013–May 2014). Box plots represent the statistical values boundaries of wind stress data for the previous week (5 days) to each sampling time. Part of the box closest and farthest to zero are the 25th and 75th percentile, respectively; horizontal lines within the box are the mean; and whiskers (error bars) are the minimum and maximum values.





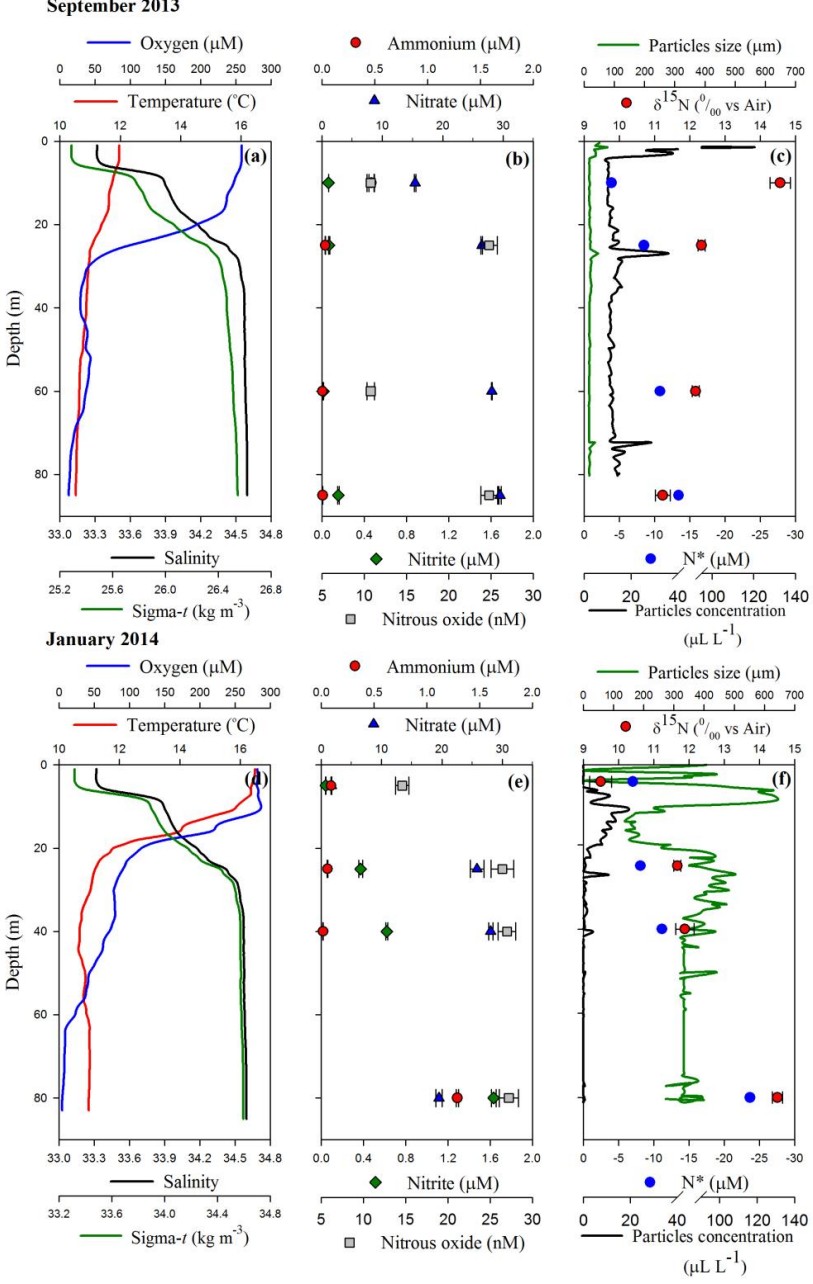

**Fig. 2**. Water column profiles of hydrographic parameters during sampling periods (a-c, 12 September 2013

5    and d-f, 28 January 2014) at Sta. 18 over the continental shelf off central Chile.





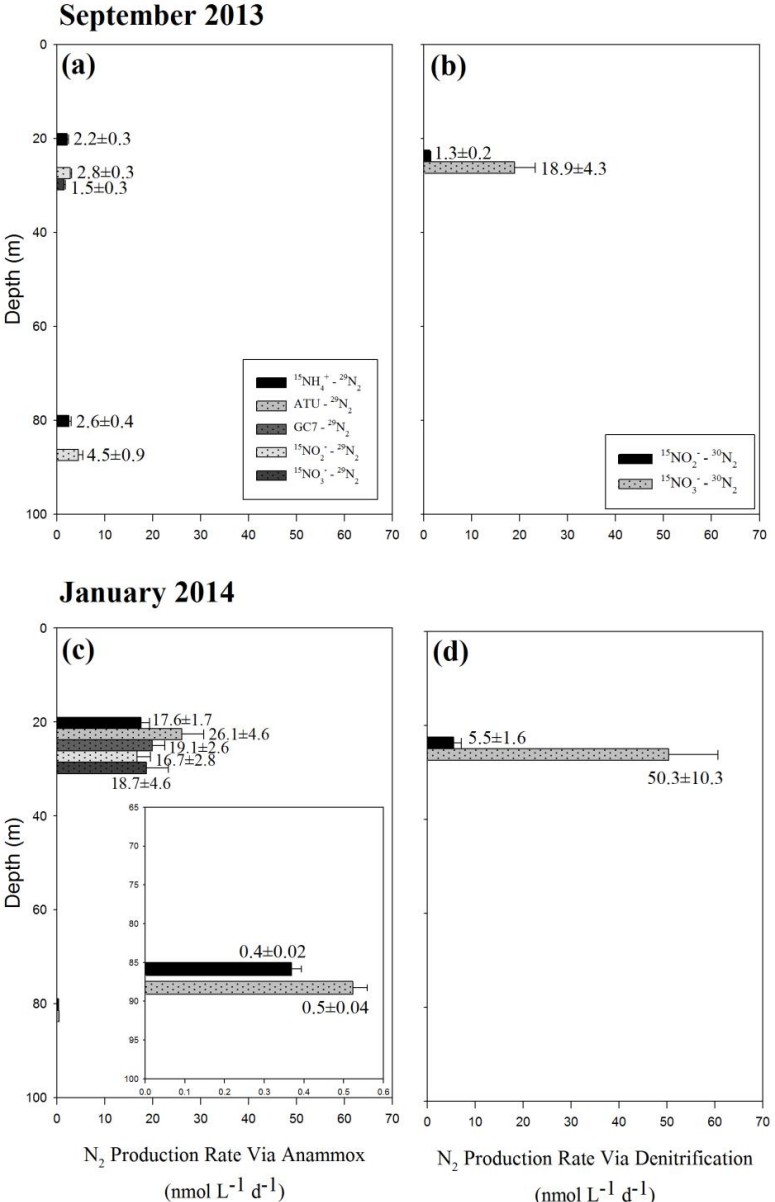

Fig. 3. Anammox (a and c) and denitrification (b and d) activities expressed as $N_2$ production rates measured in the oxycline (25 m depth) and bottom layer (85 m depth) at Sta. 18 over the continental shelf off central Chile during 12 September 2013 (upper panels) and 28 January 2014 (lower panels). Conventions show the tracer amended ($^{15}NH_4^+$, $^{15}NO_2^-$, and $^{15}NO_3^-$), isotopic pairing in the gas produced, and inhibitor when used (ATU, GC7, and acetylene – "$C_2H_2$"). Inlet in (c) represents a zoom for lower values.





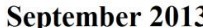

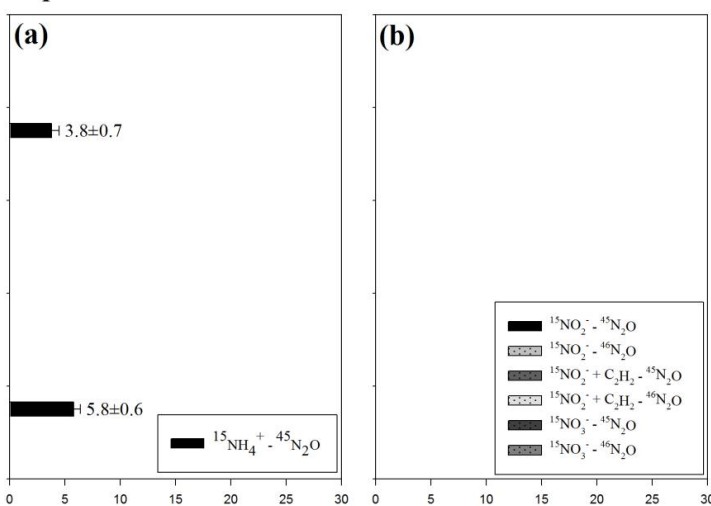

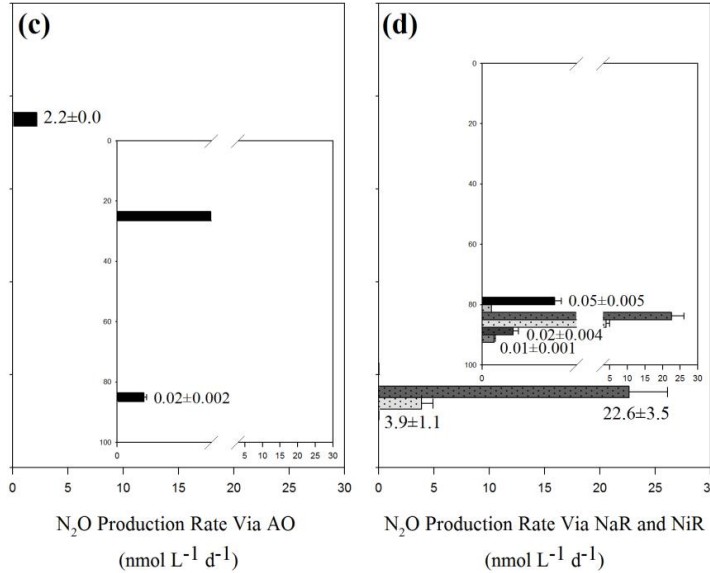

**Fig. 4**. N$_2$O production rates through AO (a and c), and NaR and NiR (b and d) measured in the oxycline (25 m depth) and bottom layer (85 m depth) at Sta. 18 over the continental shelf off central Chile during 12 September 2013 (upper panels) and 28 January 2014 (lower panels). Conventions show the tracer amended ($^{15}$NH$_4^+$, $^{15}$NO$_2^-$, and $^{15}$NO$_3^-$), isotopic pairing in the gas produced, and inhibitor when used (acetylene – "C$_2$H$_2$"). Inlets in (c) and (d) represent a zoom for lower value.



**Fig. 5**. Nitrite (a and b) and ammonium (c and d) production rates measured as $^{15}N_2$ after chemical conversion of the dissolved $^{15}N$-inorgnic compound produced (formed by redox reactions) measured in the oxycline (25 m depth) and bottom layer (85 m depth) at Sta. 18 over the continental shelf off central Chile during 12 September 2013 (upper panels) and 28 January 2014 (lower panels). Conventions denote tracer/source amended ($^{15}NH_4^+$, $^{15}NO_3^-$, and $^{15}NO_2^-$) and inhibitor (ATU and GC7) that showed activities.