# Peer review of "Vertical segregation among pathways mediating nitrogenloss ( $N_2$ and $N_2$ O production) across the oxygen gradient in a coastal upwelling ecosystem"

_Biogeosciences, 2016_

## Referee Comment (RC1) · Anonymous Referee #1 · 23 Apr 2017

The manuscript "Vertical segregation among pathways mediating nitrogen-loss (N2 and N2O production) across the oxygen gradient in a coastal upwelling ecosystem" from Galan et al. presents measurements of production rates of different nitrogen cycle processes from a station off Northern Chile along with hydrographic and stable isotope data, measured at different times during the upwelling season.

The suite of experiments conducted gives a comprehensive insight into the N cycling processes and their relative contributions to the N budget, particularly due to the combination of different 15N tracer incubations with inhibitor experiments. The manuscript

is generally well written and well-structured and includes a profound discussion of the processes investigated during the study, and I find the manuscript generally worth publishing.

However, I have some difficulties with the conclusion drawn by the authors that the temporal differences observed between the two sampling dates are mainly driven by short-term variations in the wind field and subsequent particle export that superimpose the seasonal signature of upwelling. The data shown by the authors do not fully support this hypothesis, though: as pointed out by the authors, sampling was carried out at the beginning and in the middle of the upwelling season. The observed difference in oxygen concentration, particle distribution and nitrogen cycling could also be attributed to the signals of intensified remineralization as a consequence of the progressing upwelling season. A clear indication that the particle distribution and thus the nitrogen cycling is directly linked to short-term variations in the wind field is missing. Further information (e.g. showing time-series data of chlorophyll and particle distribution along with the analysis of the wind field) should be provided to support this hypothesis.

Specific comments:

P 5, L 16: Please provide a reference for the particle size and abundance measurements.

P 5, L 26: please provide a reference for the method used for N2O measurements. What is the relative uncertainty for these measurements?

P 8, L 12: replace "hydrographic variability" with "hydrographic conditions"

P 11, L 31-34: please further explain how the particle distribution is governed by wind-induced upwelling (see my comment above).

P 13, L23: . . .can be mostly accounted for by. . .

P 13, L 21-36: are there other data on the organic matter distribution (e.g measurements of POM or DOM) available to directly assess the OM availability during the samplings or can the OM availability estimated from the particle distribution?

P 15, L 8-10: please cite also Löscher et al. (2012) for an increased N2O production from archaeal nitrification.

―――――――――――――――――

---

## Referee Comment (RC2) · Q. Ji (Referee) · 23 May 2017

This is a well-written paper, which presents results from 15N-label experiments to gain insights into a number of N-cycling processes (production of dinitrogen, nitrous oxide, nitrite and ammonium) at a coastal upwelling system. The water column oxygen of this system fluctuates seasonally at large magnitude, making this system unique. The authors also examined the seasonal and vertical differences of N-cycling rates that are ultimately dictated by the physical forcing. This dataset is worthy of publication in Biogeosciences if the authors improve the data interpretation and manuscript clarity. A

number of points are raised below.

1. Issues concerning scientific quality

One of the findings claimed by the authors is the activity of canonical denitrification and anammox in the oxygenated depths. Also, the authors detected nitrification activity under anoxic conditions (Table 3). These interesting findings may reflect the lack of knowledge about central Chilean coastal upwelling system, but they could also be the result of experimental artifacts. Upon careful examination of the oxygen concentrations during the 15N incubation experiments, the validity of these claims and results remain questionable.

The use of inhibitors is one of the highlights of this manuscript. ATU and GC-7 were used to distinguish the nitrification activity contributed from bacteria and archaea, respectively. I think the experiments were successful, as demonstrated in the results of Jan. 2014 (Figure 5b). The sum of nitrite production rates of ATU- and GC7-treatments matches (within the scale of error bars) the rate using 15-ammonium without inhibitor. So it is strange for the authors to claim that GC7 was not effective during the incubation experiment (page 15 line 30-32).

It is confusing that the use of acetylene was to quantify N2O production rate from nitrite reduction (page 7, line 11-13). More appropriately, acetylene addition in 15N-nitrite incubation is to quantify the rate of N2O consumption (equivalent to N2 production), which is the difference in N2O production rates with and without acetylene. Thus, I find it inconsistent in the denitrification data presented in figure 3 and N2O production data in figure 4. If canonical denitrification was active in the oxycline (25 m) as demonstrated in figure 3b and 3d, why was there no N2O production rate at 25 m with acetylene addition (figure 4b and 4d)? The 85 m sample from Jan. 2014 showed high N2O production rate with acetylene, and very low N2O production rate without acetylene (Figure 4d). Why was there no canonical denitrification detected at the same depth? (figure 3d).

[Figure]

In-depth interpretation of natural nitrate isotope data is needed. The author claimed that nitrate consumption (phytoplankton uptake and denitrification) increased surface delta 15N value in September. However, in January, lower surface delta 15N value (5 permil lower than September) and lower surface nitrate concentration were measured under increased denitrification rate (and potentially increased phytoplankton uptake due to higher Chl-a). The author argued that nitrogen fixation was responsible for lowering the delta 15N. If this is really the case, will nitrogen fixation be the major N source in this region, and will argue against the conclusion that upwelled nitrate is the main fuel for this system (page 17, line16 - 19)? The authors can strengthen the arguments by providing simple calculation to show how much of nitrogen fixation is needed to lower the delta 15N of nitrate by 5 permil.

2. Issues concerning presentation quality

The authors did not provide any molecular data about the microbial community throughout this manuscript. So the sentence starts in line 36 page 2 should be revised to accurately describe the data.

The naming of season and/or sampling times can be easier for readers. When referring to sampling period, which happens in a short time scale, I suggest using "September" and "January". When discussing seasonal features on a longer time scale, please use "spring" and "summer" There are some details in the "Methods" section that need to be addressed and clarified (1) Page 6, line 23. How long were the 250 ml bottles kept before processing in the laboratory? (2) Table 2, provide the in situ oxygen concentrations for all sampling dates and depths. (3) Page 6, line 30. Provide the volume of water in the 12.6 ml vials because it significantly changes the oxygen concentration in the water phase. Also, how was oxygen (and potentially nitrogen) contamination avoided during water samples transfer from 250 ml bottles to 12.6 ml vials? (4) For January, 25 m samples, it could be technically challenging to measure 15N in N2 if ambient N2 was not removed from the 12.6 ml vials. (5) Page 7, section 2.7. Provide the relative standard error for nitrate isotope analysis. (6) Page 8, line 5. For N2O production, the

total rates of N2O production is different from the sum of 45- and 46-N2O production because of different fraction labeled of substrate. Please clarify in the main text and in table 2.

The presentation style of the figures can be modified. Overall I hope the authors solve the following issues in the resubmitted manuscript. (1) Figure 1 (a), an enlarged color bar can be placed outside the Chl-a maps. Figure 1 (b), please point out the seasons on the bar plot to help readers navigate. (2) I find the figure 2 difficult to read. Firstly, the panels should be increased to 8 or 10 (4-5 panels each sampling time), so that physical and chemical parameters can be plotted on different panels without confusing x-axis. Secondly, use dashed lines to connect individual measurements to help readers navigate. Thirdly, please emphasize that the "delta 15N" refers to nitrogen isotopic composition of nitrate. (3) It is odd to use bar plots in figure 3, 4 and 5. Firstly, space is wasted when only two depths are presented, and many of the rates are low. Secondly, bar plots can be confusing because they are sometimes misplaced along the depth-axis. On fig 3c the bars are spread between 20 to 30 m but in fact they all represent rates at 25 m. Thirdly, the greyscale color scheme makes many of the bars indistinguishable even on a computer screen (e.g. figure 4d). Fourthly, the fonts of the numbers alongside the axis are different, some are very small, some are ok. In figure 4 the numbers for depth are missing. Lastly, if the value of each rate is shown, there's no need to have a separate zoom-in plot.

The discussion section 4.2 is not well organized and it makes me difficult to grasp authors' main ideas. The first 5 paragraphs (page 12 line 10 to page 14 line 15) are about N2 production. The following two paragraphs (page 14 line 17 to page 15 line 10) are for N2O production. The last two paragraphs (page 15 line 11 to 32, page 15 line 34 to page 16 line 18) are for nitrification and DNRA, respectively. I suggest breaking down section 4.2 into multiple sections (or subsections) to help readers navigate.

---

## Author Comment (AC1) · 20 Jun 2017

Anonymous Referee #1 Q:... I have some difficulties with the conclusion drawn by the authors that the temporal differences observed between the two sampling dates are mainly driven by short-term variations in the wind field and subsequent particle export that superimpose the seasonal signature of upwelling. A: Due to the lack of data on particle distributions that could match the high frequency of wind observation (daily scale) it is not possible to further support this part of the conclusion. Hence, the conclusion will instead be focused in the

influence of the upwelling season progress on the organic matter availability and its distribution, thus: P 17, L 21-32: "Considering that N removal processes are initially fueled by organic matter and the concomitant production of electron donors (e.g., NH4+) and acceptors (e.g., NO2-) after its remineralization, the vertical and temporal differences in N-loss processes during this study, and previous studies in the area (Galán et al., 2014), highlight the influence of the development of the upwelling season on the availability and vertical distribution of particulate organic matter. Furthermore, the accumulation of sinking organic particles around the oxycline considerably increases the volume for removing N processes in this coastal system. Nonetheless, and despite the great amount of regenerated ammonium produced after the oxidation of this organic load, and its central role in regulating the generation of N2O at the oxycline, the main substrate that supports the N-removal as either N2 and N2O produced, is the overabundance of preformed N, mostly in the form of nitrate, which fertilizes the system through the upwelling season. Thus, during this productive period, pulses of accumulation and consumption of different N substrates modulated the structure and activity of the microbial N-based assemblage of this coastal upwelling system"

Specific comments:

Q: P 5, L 16: Please provide a reference for the particle size and abundance measurements. A: Done: P 5, L 16: "(LISST-25X; http://www.sequoiasci.com/product/lisst-25x/)"

Q: P 5, L 26: please provide a reference for the method used for N2O measurements. What is the relative uncertainty for these measurements? A: Done: P5, L 27-28: "The coefficient of variation of the dissolved N2O measurements was < 3% (for more details, see Cornejo and Farías, 2012)."

Q: P 8, L 12: replace "hydrographic variability" with "hydrographic conditions" A: Done: P8, L 16

Q: P 11, L 31-34: please further explain how the particle distribution is governed by

wind- induced upwelling (see my comment above). A: Done: A short explanation and some references about how periods of relaxation in the wind stress (that favor the upwelling events) are necessary to allow the bloom of phytoplankton and the biomass accumulation were included (P 12, L 3-8), and the assertion about wind control over particles settling was changed and exposed as a hypothesis (and P 12, L 14).

Q: P 13, L23: . . .can be mostly accounted for by.  A: Done: P 13, L 33

Q: P 13, L 21-36: are there other data on the organic matter distribution (e.g measurements of POM or DOM) available to directly assess the OM availability during the samplings or can the OM availability estimated from the particle distribution? A: Done: Although there are no measurements of POM or DOM during the sampling periods of this study, evidence of OM accumulation through the upwelling season for this system has been reported in previous studies. These references were included: P 13 L 35-37: "Increases in the organic matter availability, measured as particulate organic carbon, during the development of the upwelling season, in both the oxycline and bottom waters have been previously reported for this coastal system (Graco et al. 2001; Galán et al., 2014)."

Q: P 15, L 8-10: please cite also Löscher et al. (2012) for an increased N2O production from archaeal nitrification. A: Done: P 15, L 22.  

Q. Ji - Referee #2

1. Issues concerning scientific quality Q: One of the findings claimed by the authors is the activity of canonical denitrification and anammox in the oxygenated depths. Also, the authors detected nitrification activity under anoxic conditions (Table 3). These interesting findings may reflect the lack of knowledge about central Chilean coastal upwelling system, but they could also be the result of experimental artifacts. Upon careful examination of the oxygen concentrations during the 15N incubation experiments, the validity of these claims and results remain questionable. A: We were also surprised to see N2 production in the oxycline waters, but the result was robust, and we discuss

possible explanations extensively (from P 12, L 27). Regarding the detection of nitrification during anoxic incubations, we note that this is a common phenomenon, which in previous studies has been attributed to the persistence of traces of oxygen in the samples (e.g., Ganesh et al. 2016, Bristow et al. 2017), although those studies exhibited similar caution to oxygen contamination as we did. We are confident in the use of the best supplies and recommendations (e.g., boiling rubber caps to remove oxygen) to minimize oxygen contamination in this kind of experiments. Furthermore, we tested the anoxia in the 250 mL bottles after 15 minutes of purging using the STOX sensor (Revsbech et al. 2009), previously to dispensing the water into the Exetainers. This information was included in Methods (P 6, L 31-32).

Q: The use of inhibitors is one of the highlights of this manuscript. ATU and GC-7 were used to distinguish the nitrification activity contributed from bacteria and archaea, respectively. I think the experiments were successful, as demonstrated in the results of Jan. 2014 (Figure 5b). The sum of nitrite production rates of ATU- and GC7-treatments matches (within the scale of error bars) the rate using 15-ammonium without inhibitor. So it is strange for the authors to claim that GC7 was not effective during the incubation experiment (page 15 line 30-32). A: This statement is because the nitrite production during the GC7 experiments was not significantly different from the control (incubations without inhibitors).

Q: It is confusing that the use of acetylene was to quantify $N_2O$ production rate from nitrite reduction (page 7, line 11-13). More appropriately, acetylene addition in 15N-nitrite incubation is to quantify the rate of $N_2O$ consumption (equivalent to $N_2$ production), which is the difference in $N_2O$ production rates with and without acetylene. . . A: It is correct that acetylene is generally applied to quantify $N_2$ production through denitrification based on the assumption that $N_2O$ is rapidly consumed. Thus, $N_2O$ production in the presence of acetylene reflects $N_2O$ consumption/$N_2$ production by denitrification. We clarified that we are looking at gross $N_2O$ production (P 7, L 17).

Q: . . .Thus, I find it inconsistent in the denitrification data presented in figure 3 and N2O

production data in figure 4. If canonical denitrification was active in the oxycline (25 m) as demonstrated in figure 3b and 3d, why was there no N2O production rate at 25 m with acetylene addition (figure 4b and 4d)? A: During September (Fig 4b) it is likely that the anoxic conditions imposed in the incubations with water from the oxycline (25 m), avoided the accumulation of N2O favoring its complete reduction to N2 (as was mentioned in the introduction (P 4, L1-5), considering that acetylene was not amended to these incubation. The absence of N2O production during January at 25 m (Fig 4d) is harder to explain. The N2 that was produced at this time and depth (Fig 3d) by canonical denitrification came mainly from assays with 15NO3 rather than 15NO2 (50.3 vs 5.5, respectively) and the assays with acetylene, performed to see N2O production, were running only for 15NO2 treatments. Lower rates of N2 production from 15NO2 implies also, in a classical view, lower N2O values. So, it is possible that these lower concentrations could not be detected during our measurements.

Q:. . . The 85 m sample from Jan. 2014 showed high N2O production rate with acety-lene, and very low N2O production rate without acetylene (Figure 4d). Why was there no canonical denitrification detected at the same depth? (figure 3d). A: We can't ex-plain this.

Q: In-depth interpretation of natural nitrate isotope data is needed. The author claimed that nitrate consumption (phytoplankton uptake and denitrification) increased surface delta 15N value in September. However, in January, lower surface delta 15N value (5 permil lower than September) and lower surface nitrate concentration were measured under increased denitrification rate (and potentially increased phytoplankton uptake due to higher Chl-a). The author argued that nitrogen fixation was responsible for lowering the delta 15N. If this is really the case, will nitrogen fixation be the major N source in this region, and will argue against the conclusion that upwelled nitrate is the main fuel for this system (page 17, line16 - 19)? The authors can strengthen the arguments by providing simple calculation to show how much of nitrogen fixation is needed to lower the delta 15N of nitrate by 5 permil. A: If the starting point is 14.5‰ (as

in surface during September; Figure 2c), and assuming that N fixation adds fixed N at 0‰ and N from fixation is converted to nitrate without fractionation (all NH4+ oxidized), it would need to add 34% of the nitrate through fixation. With nitrate depleted to 2 $\mu$M (as in surface during January; Figure 2e), this corresponds to the adition of just 0.7 $\mu$M fixed N. It would be less if the starting point is less positive. So based on this, the input from fixation is not necessarily so big, but obvisously more details are needed.

2. Issues concerning presentation quality Q: The authors did not provide any molecular data about the microbial community through- out this manuscript. So the sentence starts in line 36 page 2 should be revised to accurately describe the data. A: We cannot find a line 36 in page 2. However, although the manuscript does not present molecular results, activities related with some of the processes measured here are discussed in the context of molecular information reported for this system.

Q: The naming of season and/or sampling times can be easier for readers. When referring to sampling period, which happens in a short time scale, I suggest using "September" and "January". When discussing seasonal features on a longer time scale, please use "spring" and "summer" A: Done

There are some details in the "Methods" section that need to be addressed and clarified

(1) Page 6, line 23. How long were the 250 ml bottles kept before processing in the laboratory? A: Done: P 6, L 25.

(2) Table 2, provide the in situ oxygen concentrations for all sampling dates and depths. A: Done. P 30.

(3) Page 6, line 30. Provide the volume of water in the 12.6 ml vials because it significantly changes the oxygen concentration in the water phase. Also, how was oxygen (and potentially nitrogen) contamination avoided during water samples transfer from 250 ml bottles to 12.6 ml vials? A: Done: P 6, L 33-34. The Exetainers were incubated without a headspace, so degassing of oxygen during incubation was not an issue.

(4) For January, 25 m samples, it could be technically challenging to measure 15N in N2 if ambient N2 was not removed from the 12.6 ml vials. A: It is correct that the background of N2 dissolved in the water will affect the sensitivity of the 15N analysis, but the effect is not large, and we regularly detect 15N2 production in unpurged water samples.

(5) Page 7, section 2.7. Provide the relative standard error for nitrate isotope analysis. A: Done: P 7, L 34

(6) Page 8, line 5. For N2O production, the total rates of N2O production is different from the sum of 45- and 46-N2O production because of different fraction labeled of substrate. Please clarify in the main text and in table 2. A: Production of 45N2O and 46N2O are reported separately because different pathways may contribute to these production rates through different patterns of isotope combination, as was established in the text (P 8, L 8-10).

Q: The presentation style of the figures can be modified. Overall I hope the authors solve the following issues in the resubmitted manuscript. (1) Figure 1 (a), an enlarged color bar can be placed outside the Chl-a maps. Figure 1 (b), please point out the seasons on the bar plot to help readers navigate. (2) I find the figure 2 difficult to read. Firstly, the panels should be increased to 8 or 10 (4-5 panels each sampling time), so that physical and chemical parameters can be plotted on different panels without confusing x-axis. Secondly, use dashed lines to connect individual measurements to help readers navigate. Thirdly, please emphasize that the "delta 15N" refers to nitrogen isotopic composition of nitrate. (3) It is odd to use bar plots in figure 3, 4 and 5. Firstly, space is wasted when only two depths are presented, and many of the rates are low. Secondly, bar plots can be confusing because they are sometimes misplaced along the depth-axis. On fig 3c the bars are spread between 20 to 30 m but in fact they all represent rates at 25 m. Thirdly, the greyscale color scheme makes many of the bars indistinguishable even on a computer screen (e.g. figure 4d). Fourthly, the fonts of the numbers alongside the axis are different, some are very small, some are ok. In figure

4 the numbers for depth are missing. Lastly, if the value of each rate is shown, there's no need to have a separate zoom-in plot. A: These issues will be solved following the editor's suggestions

Q: The discussion section 4.2 is not well organized and it makes me difficult to grasp authors' main ideas. The first 5 paragraphs (page 12 line 10 to page 14 line 15) are about N2 production. The following two paragraphs (page 14 line 17 to page 15 line 10) are for N2O production. The last two paragraphs (page 15 line 11 to 32, page 15 line 34 to page 16 line 18) are for nitrification and DNRA, respectively. I suggest breaking down section 4.2 into multiple sections (or subsections) to help readers navigate. A: Done.

---

## Author Response (AR1)

Due to the lack of data on particle distributions that could match the high frequency of wind observation (daily scale) it is not possible to further support this part of the conclusion. Hence, the conclusion will instead be focused in the influence of the upwelling season progress on the organic matter availability and its distribution, thus:

*P 17, L 21-32: "Considering that N removal processes are initially fueled by organic matter and the concomitant production of electron donors (e.g., NH4+) and acceptors (e.g., NO2-) after its remineralization, the vertical and temporal differences in N-loss processes during this study, and previous studies in the area (Galán et al., 2014), highlight the influence of the development of the upwelling season on the availability and vertical distribution of particulate organic matter. Furthermore, the accumulation of sinking organic particles around the oxycline considerably increases the volume for removing N processes in this coastal system. Nonetheless, and despite the great amount of regenerated ammonium produced after the oxidation of this organic load, and its central role in regulating the generation of N$_2$O at the oxycline, the main substrate that supports the N-removal as either N$_2$ and N$_2$O produced, is the overabundance of preformed N, mostly in the form of nitrate, which fertilizes the system through the upwelling season. Thus, during this productive period, pulses of accumulation and consumption of different N substrates modulated the structure and activity of the microbial N-based assemblage of this coastal upwelling system."*

Specific comments:

P 5, L 16: Please provide a reference for the particle size and abundance measurements.

Done: P 5, L 16: *"(LISST-25X; http://www.sequoiasci.com/product/lisst-25x/)"*

P 5, L 26: please provide a reference for the method used for N2O measurements. What is the relative uncertainty for these measurements?

Done: P5, L 27-28: *"The coefficient of variation of the dissolved N$_2$O measurements was < 3% (for more details, see Cornejo and Farías, 2012)."*

P 8, L 12: replace "hydrographic variability" with "hydrographic conditions"

Done: P8, L 16

P 11, L 31-34: please further explain how the particle distribution is governed by wind-induced upwelling (see my comment above).

Done: A short explanation and some references about how periods of relaxation in the wind stress (that favor the upwelling events) are necessary to allow the bloom of phytoplankton and the biomass accumulation were included (P 12, L 3-8), and the assertion about wind control over particles settling was changed and exposed as a hypothesis (and P 12, L 14).

P 13, L23: . . .can be mostly accounted for by. . .

Done: P 13, L 33

P 13, L 21-36: are there other data on the organic matter distribution (e.g measurements of POM or DOM) available to directly assess the OM availability during the samplings or can the OM availability estimated from the particle distribution?

Done: Although there are no measurements of POM or DOM during the sampling periods of this study, evidence of OM accumulation through the upwelling season for this system has been reported in previous studies. These references were included:

P 13 L 35-37: "*Increases in the organic matter availability, measured as particulate organic carbon, during the development of the upwelling season, in both the oxycline and bottom waters have been previously reported for this coastal system (Graco et al. 2001; Galán et al., 2014).*"

P 15, L 8-10: please cite also Löscher et al. (2012) for an increased N2O production from archaeal nitrification.

Done: P 15, L 22

**Q. Ji - Referee #2**

1. Issues concerning scientific quality

One of the findings claimed by the authors is the activity of canonical denitrification and anammox in the oxygenated depths. Also, the authors detected nitrification activity under anoxic conditions (Table 3). These interesting findings may reflect the lack of knowledge about central Chilean coastal upwelling system, but they could also be the result of experimental artifacts. Upon careful examination of the oxygen concentrations during the 15N incubation experiments, the validity of these claims and results remain questionable.

We were also surprised to see $N_2$ production in the oxycline waters, but the result was robust, and we discuss possible explanations extensively (from P 12, L 27). Regarding the detection of nitrification during anoxic incubations, we note that this is a common phenomenon, which in previous studies has been attributed to the persistence of traces of oxygen in the samples (e.g., Ganesh et al. 2016, Bristow et al. 2017), although those studies exhibited similar caution to oxygen contamination as we did. We are confident in the use of the best supplies and recommendations (e.g., boiling rubber caps to remove oxygen) to minimize oxygen contamination in this kind of experiments. Furthermore, we tested the anoxia in the 250 mL bottles after 15 minutes of purging using the STOX sensor (Revsbech et al. 2009), previously to dispensing the water into the Exetainers. This information was included in Methods (P 6, L 31-32).

The use of inhibitors is one of the highlights of this manuscript. ATU and GC-7 were used to distinguish the nitrification activity contributed from bacteria and archaea, respectively. I think the experiments were successful, as demonstrated in the results of Jan. 2014 (Figure 5b). The sum of nitrite production rates of ATU- and GC7-treatments matches (within the scale of error bars) the rate using 15-ammonium without inhibitor. So it is strange for the authors to claim that GC7 was not effective during the incubation experiment (page 15 line 30-32).

This statement is because the nitrite production during the GC7 experiments was not significantly different from the control (incubations without inhibitors).

It is confusing that the use of acetylene was to quantify $N_2O$ production rate from nitrite reduction (page 7, line 11-13). More appropriately, acetylene addition in [15]N-nitrite incubation is to quantify the rate of $N_2O$ consumption (equivalent to $N_2$ production), which is the difference in $N_2O$ production rates with and without acetylene…

It is correct that acetylene is generally applied to quantify $N_2$ production through denitrification based on the assumption that $N_2O$ is rapidly consumed. Thus, $N_2O$ production in the presence of acetylene reflects $N_2O$ consumption/$N_2$ production by denitrification. We clarified that we are looking at gross $N_2O$ production (P 7, L 17).

…Thus, I find it inconsistent in the denitrification data presented in figure 3 and $N_2O$ production data in figure 4. If canonical denitrification was active in the oxycline (25 m) as demonstrated in figure 3b and 3d, why was there no $N_2O$ production rate at 25 m with acetylene addition (figure 4b and 4d)?

During September (Fig 4b) it is likely that the anoxic conditions imposed in the incubations with water from the oxycline (25 m), avoided the accumulation of $N_2O$ favoring its complete reduction to $N_2$ (as was mentioned in the introduction (P 4, L1-5), considering that acetylene was not amended to these incubation.

The absence of $N_2O$ production during January at 25 m (Fig 4d) is harder to explain. The $N_2$ that was produced at this time and depth (Fig 3d) by canonical denitrification came mainly from assays with $^{15}NO_3$ rather than $^{15}NO_2$ (50.3 vs 5.5, respectively) and the assays with acetylene, performed to see $N_2O$ production, were running only for $^{15}NO_2$ treatments. Lower rates of $N_2$ production from $^{15}NO_2$ implies also, in a classical view, lower $N_2O$ values. So, it is possible that these lower concentrations could not be detected during our measurements.

… The 85 m sample from Jan. 2014 showed high $N_2O$ production rate with acetylene, and very low $N_2O$ production rate without acetylene (Figure 4d). Why was there no canonical denitrification detected at the same depth? (figure 3d).

We can't explain this.

In-depth interpretation of natural nitrate isotope data is needed. The author claimed that nitrate consumption (phytoplankton uptake and denitrification) increased surface delta $^{15}N$ value in September. However, in January, lower surface delta $^{15}N$ value (5 permil lower than September) and lower surface nitrate concentration were measured under increased denitrification rate (and potentially increased phytoplankton uptake due to higher Chl-a). The author argued that nitrogen fixation was responsible for lowering the delta $^{15}N$. If this is really the case, will nitrogen fixation be the major N source in this region, and will argue against the conclusion that upwelled nitrate is the main fuel for this system (page 17, line16 - 19)? The authors can strengthen the arguments by providing simple calculation to show how much of nitrogen fixation is needed to lower the delta $^{15}N$ of nitrate by 5 permil.

If the starting point is 14.5‰ (as in surface during September; Figure 2c), and assuming that N fixation adds fixed N at 0‰ and N from fixation is converted to nitrate without fractionation (all $NH_4^+$ oxidized), it would need to add 34% of the nitrate through fixation. With nitrate depleted to 2 µM (as in surface during January; Figure 2e), this corresponds to the addition of just 0.7 µM fixed N. It would be less if the starting point is less positive. So based on this, the input from fixation is not necessarily so big, but obviously more details are needed.

2. Issues concerning presentation quality

The authors did not provide any molecular data about the microbial community through-out this manuscript. So the sentence starts in line 36 page 2 should be revised to accurately describe the data.

We cannot find a line 36 in page 2. However, although the manuscript does not present molecular results, activities related with some of the processes measured here are discussed in the context of molecular information reported for this system.

The naming of season and/or sampling times can be easier for readers. When referring to sampling period, which happens in a short time scale, I suggest using "September" and "January". When discussing seasonal features on a longer time scale, please use "spring" and "summer"

Done.

There are some details in the "Methods" section that need to be addressed and clarified (1) Page 6, line 23. How long were the 250 ml bottles kept before processing in the laboratory?

Done: P 6, L 25.

(2) Table 2, provide the in situ oxygen concentrations for all sampling dates and depths.

Done. P 30-32.

(3) Page 6, line 30. Provide the volume of water in the 12.6 ml vials because it significantly changes the oxygen concentration in the water phase. Also, how was oxygen (and potentially nitrogen) contamination avoided during water samples transfer from 250 ml bottles to 12.6 ml vials?

Done: P 6, L 33-34. The Exetainers were incubated without a headspace, so degassing of oxygen during incubation was not an issue.

(4) For January, 25 m samples, it could be technically challenging to measure $^{15}$N in $N_2$ if ambient $N_2$ was not removed from the 12.6 ml vials.

It is correct that the background of $N_2$ dissolved in the water will affect the sensitivity of the $^{15}$N analysis, but the effect is not large, and we regularly detect $^{15}N_2$ production in unpurged water samples.

(5) Page 7, section 2.7. Provide the relative standard error for nitrate isotope analysis.

Done: P 7, L 34

(6) Page 8, line 5. For $N_2O$ production, the total rates of $N_2O$ production is different from the sum of $^{45}$- and $^{46}$-N2O production because of different fraction labeled of substrate. Please clarify in the main text and in table 2.

Production of $^{45}N_2O$ and $^{46}N_2O$ are reported separately because different pathways may contribute to these production rates through different patterns of isotope combination, as was established in the text (P 8, L 8-10).

The presentation style of the figures can be modified. Overall I hope the authors solve the following issues in the resubmitted manuscript. (1) Figure 1 (a), an enlarged color bar can

be placed outside the Chl-a maps.

Done.

Figure 1 (b), please point out the seasons on the bar plot to help readers navigate.

Done. A shaded area was included to denote the upwelling season.

(2) I find the figure 2 difficult to read. Firstly, the panels should be increased to 8 or 10 (4-5 panels each sampling time), so that physical and chemical parameters can be plotted on different panels without confusing x-axis. Secondly, use dashed lines to connect individual measurements to help readers navigate. Thirdly, please emphasize that the "delta 15N" refers to nitrogen isotopic composition of nitrate.

Done. However, dashed lines were not included because this assumes an average behavior between the two connected points.

(3) It is odd to use bar plots in figure 3, 4 and 5. Firstly, space is wasted when only two depths are presented, and many of the rates are low.

Done. Figure size was reduced.

Secondly, bar plots can be confusing because they are sometimes misplaced along the depth-axis. On fig 3c the bars are spread between 20 to 30 m but in fact they all represent rates at 25 m.

Done. Just the two sampling depths (25 and 85 m) were left on the y-axis.

Thirdly, the greyscale color scheme makes many of the bars indistinguishable even on a computer screen (e.g. figure 4d).

Done. Color of bars was changed in order to facilitate differentiation among treatments.

Fourthly, the fonts of the numbers alongside the axis are different, some are very small, some are ok. In figure 4 the numbers for depth are missing.

Done. Fonts were unified.

Lastly, if the value of each rate is shown, there's no need to have a separate zoom-in plot.

Inlets that show zooms were maintained, otherwise the lower rates are hard or impossible to see.

The discussion section 4.2 is not well organized and it makes me difficult to grasp authors' main ideas. The first 5 paragraphs (page 12 line 10 to page 14 line 15) are about N2 production. The following two paragraphs (page 14 line 17 to page 15 line 10) are for N2O production. The last two paragraphs (page 15 line 11 to 32, page 15 line 34 to page 16 line 18) are for nitrification and DNRA, respectively. I suggest breaking down section 4.2 into multiple sections (or subsections) to help readers navigate.

Done.

---

## Referee Report (RR1)

**Referee report to version 2.**

Overall this is a major improvement since the last draft. I recommend acceptance after minor revision.

After reviewing the revised manuscript and the authors' response, I think there is one major problem that the authors should clarify: The inconsistency between N2 production and gross N2O production with acetylene, which is demonstrated in figure 3d and 4d. Current understanding is that, N2O is an intermediate in the sequence of denitrification. When N2 production is measured by nitrite reduction, N2O production with acetylene should have similar rates. This was not observed in the result; quite surprisingly, N2 production and N2O production decoupled. The authors can add some comments in line 30, page 10.

Three minor issues concerning the nitrite production experiments that I would hope the authors clarify.

First, in section 3.3.3 (page 10-11), the authors claimed major sources of nitrite as either nitrate reduction or ammonium oxidation in September sampling. Drawing this conclusion should be careful because the oxygen condition was manipulated, at least for 25 m samples, in the incubation experiment.

Second, the authors should point out that ammonium oxidation to nitrite occurred under helium-purged samples, in section 3.3.3 and section 4.2.3. And use some of their 'response to reviewers' in the main text to support their experimental findings (i.e. high oxygen affinity during ammonium oxidation in cultures and natural waters). In page 6, line 31 – 34 the authors stated the procedure of the incubation. Unless the authors transferred the water in oxygen-free environment (e.g. glove bag or anaerobic chamber), oxygen contamination is highly likely. I think it more important to validate the oxygen concentration in the 12 ml vials, rather than the 250 ml bottle.

Third, I would encourage the authors to acknowledge their GC7 and ATU inhibitor experiments provided internally consistent data, and both inhibitors were working well (P14, L6 – 8). As I have stated earlier, the sum of nitrite production rates of ATU- and GC7-treatments matches, within the scale of error bars, the rate using 15-ammonium without inhibitor. Because ATU experiments yielded very low rates, nitrite production during the GC7 experiment was close to the rate from the control. If GC7 was not an effective inhibitor, then ATU would not be effective as well, and the experimental results would become a big question mark. I think the central Chilean upwelling system is unique in that bacterial contribution to ammonium oxidation is more significant because of influence from shelf sediment. For this case, it is not an issue that the authors found different things than the other investigators. It is an issue treating their own data unfairly.

Other wording issues:

P2, L 2-3: Confusing sentence. Do you mean the oxycline, from oxygen saturation to anoxia, spans only a few meters, located in 30 – 50 meters depth?

P2, L4: Nitrification is not considered as nitrogen removal process. Please revise accordingly

P2, L15: What do you mean by 'main subsidiary processes'? Wordiness and confusing. Also, define DNRA because in the main text there are terms like 'DNaRA' and 'DNiRA'. Please make it easier to the readers. I would recommend using the full name of these processes.

P4, L3: Delete 'mechanism or'

P4, L4: What is 'almost completely'? Wordiness and confusing. Under anoxia (no oxygen), $N_2O$ is completely reduced to $N_2$.

P6, L30: Change to 'purged with He for 15 min'

P10, L25: I would choose 'September' rather than 'spring' here. I request once again about naming of the seasons and sampling times throughout the text. When referring to sampling period, which happens in a short time scale, I suggest using "September" and "January". When discussing seasonal features on a longer time scale, please use "spring" and "summer".

P14, L6: Please change: 'related to'.

P17, L11 – 15: Here, I would suggest adding some simple calculation, just as they showed in the 'response to reviewers', to demonstrate the N fixation is a source of nitrate that should not be neglected.

Caption in figure 4: State that the $N_2O$ production rate when measured under $C_2H_2$ addition, is 'gross production rate'.

---

## Author Response (AR2)

Although we expected to obtain similar rates of $N_2$ and the $N_2O$ production by nitrite reduction without and with acetylene, respectively (Figs., 3d and 4d), a mismatch between these two products was observed that we cannot explain as denitrification is the only mechanism known to produce $N_2O$ in the presence of acetylene.

Three minor issues concerning the nitrite production experiments that I would hope the authors clarify.

First, in section 3.3.3 (page 10-11), the authors claimed major sources of nitrite as either nitrate reduction or ammonium oxidation in September sampling. Drawing this conclusion should be careful because the oxygen condition was manipulated, at least for 25 m samples, in the incubation experiment.

Done. P11, L5-6. A note on the differences between oxygen conditions during the incubations with samples from the oxycline was included.

Second, the authors should point out that ammonium oxidation to nitrite occurred under helium-purged samples, in section 3.3.3 and section 4.2.3…

Done. P11, L5-6. A note on the differences between oxygen conditions during the incubations with samples from the oxycline was included.

…And use some of their 'response to reviewers' in the main text to support their experimental findings (i.e. high oxygen affinity during ammonium oxidation in cultures and natural waters). In page 6, line 31 – 34 the authors stated the procedure of the incubation. Unless the authors transferred the water in oxygen-free environment (e.g. glove bag or anaerobic chamber), oxygen contamination is highly likely. I think it more important to validate the oxygen concentration in the 12 ml vials, rather than the 250 ml bottle.

Done. P13, L31-34. A paragraph about the detection of nitrification during anoxic incubations was included.

Third, I would encourage the authors to acknowledge their GC7 and ATU inhibitor experiments provided internally consistent data, and both inhibitors were working well (P14, L6 – 8). As I have stated earlier, the sum of nitrite production rates of ATU- and GC7-treatments matches, within the scale of error bars, the rate using 15-ammonium without inhibitor. Because ATU experiments yielded very low rates, nitrite production during the GC7 experiment was close to the rate from the control. If GC7 was not an effective inhibitor, then ATU would not be effective as well, and the experimental results would become a big question mark. I think the central Chilean upwelling system is unique in that bacterial contribution to ammonium oxidation is more significant because of influence from shelf sediment. For this case, it is not an issue that the authors found different things than the other investigators. It is an issue treating their own data unfairly.

Based on the perspective presented by the Referee here, it is true that the sum of the nitrite rates from the ATU and CG7 treatments is close to the rate found in the control treatment. However, these experiments were developed in different incubations; so, the correct way to interpret the results is to compare each inhibitor treatment with the control separately. In this case, the GC7 results were not significantly different from the control. Therefore, at the beginning of the paragraph (P. 15, L 28-29), it was stated that bacteria play a major role modulating the ammonium oxidation relative to archaea does. Nevertheless, considering the biogeochemical and molecular background reported in this system we propose the possibly that GC7 could not affect archaeal metabolism in incubations of short duration. Then, no changes were made it in the document.

Other wording issues:

P2, L 2-3: Confusing sentence. Do you mean the oxycline, from oxygen saturation to anoxia, spans only a few meters, located in 30 – 50 meters depth?

Done. P2, L2. Sentence was modified.

P2, L4: Nitrification is not considered as nitrogen removal process. Please revise accordingly

This is true in the classical view, but in the context of lower oxygen conditions this process participates in the nitrogen loss due the reaction of ammonium oxidation could produce $N_2O$ as will be demonstrated through the manuscript. In this sense, no changes were made it in the document.

P2, L15: What do you mean by 'main subsidiary processes'? Wordiness and confusing.

Done. P2, L15. Sentence was modified.

Also, define DNRA because in the main text there are terms like 'DNaRA' and 'DNiRA'. Please make it easier to the readers. I would recommend using the full name of these processes.

Done. P2, L15, and throughout the text.

P4, L3: Delete 'mechanism or'

Done. P4, L3.

P4, L4: What is 'almost completely'? Wordiness and confusing. Under anoxia (no oxygen), N2O is completely reduced to N2.

Done. P4, L4. Sentence was modified.

P6, L30: Change to 'purged with He for 15 min'

Done. P6, L30.

P10, L25: I would choose 'September' rather than 'spring' here. I request once again about naming of the seasons and sampling times throughout the text. When referring to sampling period, which happens in a short time scale, I suggest using "September" and "January". When discussing seasonal features on a longer time scale, please use "spring" and "summer".

Done. P10, L25, and throughout the text.

P14, L6: Please change: 'related to'.

Done. P14, L19.

P17, L11 – 15: Here, I would suggest adding some simple calculation, just as they showed in the 'response to reviewers', to demonstrate the N fixation is a source of nitrate that should not be neglected.

Done. P17, L19-24. A sentence about it was included.

Caption in figure 4: State that the N2O production rate when measured under C2H2 addition, is 'gross production rate'.

Done. P36, L7. An annotation was included.